# Transferring structural knowledge across cognitive maps in humans and models

Shirley Mark [1]✉, Rani Moran [2], Thomas Parr[1], Steve W. Kennerley [3] & Timothy E. J. Behrens[4,5]

Relations between task elements often follow hidden underlying structural forms such as periodicities or hierarchies, whose inferences fosters performance. However, transferring structural knowledge to novel environments requires flexible representations that are generalizable over particularities of the current environment, such as its stimuli and size. We suggest that humans represent structural forms as abstract basis sets and that in novel tasks, the structural form is inferred and the relevant basis set is transferred. Using a computational model, we show that such representation allows inference of the underlying structural form, important task states, effective behavioural policies and the existence of unobserved state-trajectories. In two experiments, participants learned three abstract graphs during two successive days. We tested how structural knowledge acquired on Day-1 affected Day-2 performance. In line with our model, participants who had a correct structural prior were able to infer the existence of unobserved state-trajectories and appropriate behavioural policies.

[1] Wellcome Trust Centre for Neuroimaging, UCL. Queen Square 12, London WC1N 3BG, UK. [2] Max Planck UCL Center for Computational Psychiatry and Aging Research, Russell Square 10-12, London WC1B 5EH, UK. [3] Sobell Department of Motor Neuroscience, University College London, London, UK. [4] Wellcome Centre for Integrative Neuroimaging, Centre for Functional Magnetic Resonance Imaging of the Brain, University of Oxford, John Radcliffe Hospital, Oxford OX3 9DU, UK. [5] Wellcome Centre for Human Neuroimaging, Institute of Neurology, University College London, 12 Queen Square, London WC1N 3BG, UK. ✉email: markshir@gmail.com

Decisions in a new environment require the understanding of what are the relevant components in this environment and how they are related to each other. In the cognitive literature, the representation that holds such information is termed a 'cognitive map'[1]. Equipped with a 'cognitive map', an animal can predict the consequence of events and actions to inform its decisions. Associative learning results in a cognitive map in which the relations between the components are encoded using the associations between the representations of the components themselves[2,3], this type of representation does not incorporate prior knowledge and cannot be generalized. In novel environments, while learning a new cognitive map, it should be beneficial to exploit relevant information that was acquired in the past. What information is relevant to transfer and how it is represented remains an open question.

One possibility is that, while sensory information may differ radically between different situations, the brain may take advantage of previously learnt structural knowledge or schemas[4–6]. Relationships between elements in different environments often follow stereotypical patterns[7–9]. Social networks, for example, are organized in communities[10]. The day–night cycle, the cycle over the seasons and the appearance of the moon in the sky all follow a periodic pattern. Hierarchies are also abundant; for example, a family, management chain in a workplace or concept organization[11]. Representing such structures confers theoretical advantages in learning when encountering a new environment. Inferring the relevant structure enables the use of policies that are beneficial in environments with a particular underlying structure. Further, relationships that have never been observed can be inferred because the structure of the problem is familiar[12–15].

One structure which is pervasive in life and which we know facilitates such inferences is the 2-dimensional topology of space. This structure can be used, for example, to infer the correct trajectory to a goal even when the intermediate locations have never been experienced (as with refs. [16,17]). If such structural knowledge can be transferred from one set of sensory events to another, it should be represented in a way that is disentangled from the sensory stimuli and the particularities of the current task. We can consider the setting in which all tasks are represented as graphs, whereby each node on the graph is a particular sensory stimulus that is currently experienced, for example, observing the shape of the moon. Then, an edge between two sensory stimuli implies a transition between sensory states; a round moon will be followed by an elliptic moon. These graphs can have different structural forms[18,19]. The lunar graph, the seasonal graph and the day–night graph will all be circular; a workplace graph will be hierarchical; the social network graph will have a community structure; and the spatial environment will have a transition structure that respects the translational and rotational invariances of 2D space. Can humans extract such abstract information and use it to facilitate new inferences? If so, how can this knowledge be represented efficiently by the brain?

Here we show that humans extract and transfer structural regularities in graph-learning tasks. When observing a new sensory environment with a familiar structural form, they infer the existence of paths they have never seen that conform to the structural form and make novel choices that are likely beneficial. In order to understand these effects, we compared different computational models. Cognitive maps can be represented using associative learning[2,3], in such a representation, relationships between the elements in the environment are encoded by the associations between the representations of the elements themselves. Therefore, the knowledge of the structure of the environment is conjugated with the stimulus representations and there is no abstraction. Using the Successor Representation (SR) as an associative model for a cognitive map[2,3], we show that such a

model cannot account for the ability of humans to infer paths that have never been observed. Smoothing of such a representation allows inference of paths that have not yet been observed[2], but such smoothing does not depend on prior knowledge. However, our participants' success on the task depends on the structure of the graph they have experienced a day before, which therefore implies that they do exploit prior structural knowledge.

To account for the exploitation of prior structural knowledge we suggest a computational mechanism for representing, inferring and transferring this abstract structural knowledge. Such a representation should allow inference of the currently relevant structural form and the transfer of relevant knowledge to new sensory environments[9,18]. Structural abstraction is inherent to common computational frameworks such as Hidden Markov Models (HMMs). However, for flexible generalisation to new environments, the representation should highlight key statistical properties of the graph structure but suppress environment-specific particularities. We show that this can be achieved by representing structural knowledge in the form of basis sets (a set of vectors that can be used for function approximation), as has been proposed in reinforcement learning[13,20]. This complements generative modelling approaches that attempt to infer low-dimensional latent states that explain high dimensional observations[21], and complies with the Bayesian Occam's razor in finding the simplest explanation for these[22]. To support our suggestion, we conducted a second experiment. We show that prior over hidden structure changes participants' strategy for learning a new graph, to match previously experienced graph statistics. Moreover, participants adjust their behavioural policy according to the hidden underlying structure.

## Results

**Task design.** We created a task in which simulated agents and humans learn abstract graphs (Fig. 1). The graphs belong to two different structural forms. Each structural form is controlled by a different connectivity rule (Fig. 1a). We focused on two structural forms, a graph with a transition matrix that obeys translational and rotational invariant symmetry (Hexagonal graph) and graphs that have underlying community structure (Fig. 1a). Each node in the graph corresponds to a sensory stimulus (a picture—which is a state of the task). Each edge implies that these sensory stimuli can appear one after the other (direct bidirectional transitions between the states are allowed[23,24]). Using this task, we asked whether our agents and participants can infer the structural form of the underlying graph and how the structural knowledge can be exploited to better accomplish the task. Participants performed the task during 2 successive days. We asked whether participants can infer the underlying structural form encountered during the first day and transfer and exploit this knowledge during the second day. On the first day, participants learned two different graphs, with different pictures set but same structure. On the second day participants learned a third graph with a new pictures set (one group learned a graph with the same underlying structure and the other group learned a graph with a different structure, Fig. 1).

The agents and participants learned the graphs by observing pairs of stimuli that are connected by an edge. Each block of the task begins with a learning phase. Following the learning phase, we examined agents' and participants' knowledge of the graph. To examine participants' knowledge of the graph, we performed four separate tests within each block of the task (Fig. 1b): (1) We asked participants to report which of two pictures sequences could be extended with a target picture. (2) Participants reported whether a target picture could appear between two other pictures in a sequence. (3) Participants navigated on the graph; starting from a

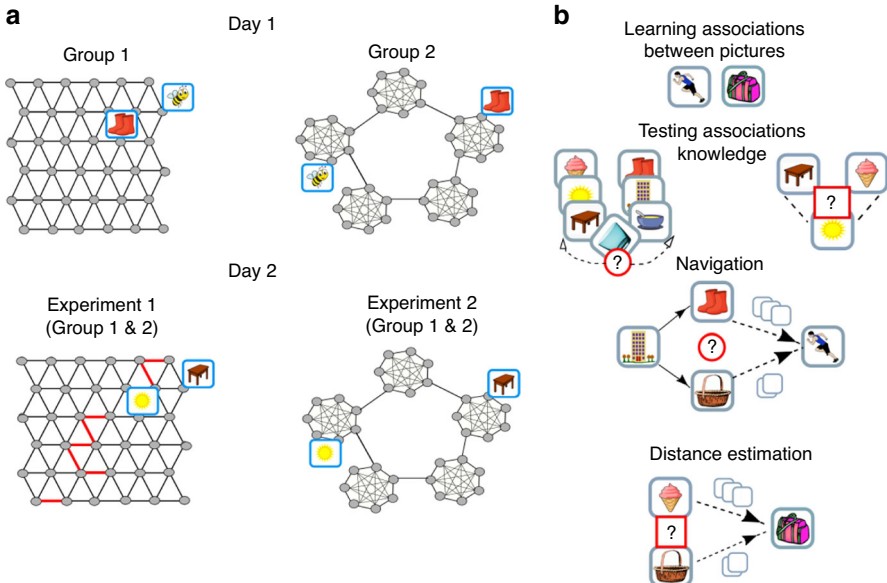

**Fig. 1 Transfer of structural knowledge: graph structures and experimental design. a** Experimental design. Agents and participants learned graphs with underlying Hexagonal (left) or Community (right) structure. Each grey dot is a node on a graph and corresponds to a picture that was viewed by the participant (for example, a picture of a bee). The lines are edges between nodes. Pictures of nodes that are connected by an edge can appear one after the other. The degree of all nodes in both graphs is six (a connecting node connects to one fewer node within a community to keep the degree equal to six). Participants learned the graphs during two successive days. In both experiments, participants were segregated into two groups. Participants in one group learned graphs with the same underlying structure during both days while the other groups learned graphs with different underlying structures during the different days. Two graphs were learnt during day 1 and additional graph on day 2. **b** One block of the task. Participants never observed the underlying graph structure but had to learn (or infer) it by performing a task. In each block, participants learned the associations between pairs of pictures, where each pair of pictures are emitted by neighbouring states on the graph. Following the learning phase, participants had to answer different types of questions: (1) report which of the two picture sequences could be extended with a target picture, (2) indicate whether the picture in the middle (sun) can appear between the two other pictures in a sequence (left and right, respectively, under 'Testing associations knowledge') and (3) they navigated on the graph: starting from a certain picture, for example, the building, participants had to choose (or skip) between two pictures that are connected to the current picture (the building) on the graph, for example, basket and boots (empty squares above the arrows indicate minimum steps to the target). The chosen picture then replaces the 'starting picture'. Participants repeated these steps until they reached the target picture (for instance, the running man). (4) Which picture is closer to the target picture (the bag), the ice-cream or the basket? (Distance estimation). Question type three was excluded from day 2 on experiment 1.

source picture, participants repeatedly chose between two of the picture's neighbours until reaching the target, with the aim to do so in the smallest number of steps. (4) Participants were asked to report which of two pictures is closer to a target picture (without feedback, see 'Methods' for further details).

We tested transfer of structural knowledge by conducting two different experiments. In each experiment, we tested the effect of transfer of a specific structural form; in the first experiment, we have tested transfer of Hexagonal grid structure, while in the second we have tested the transfer of community-structure knowledge (see below for more details). In each experiment, participants learned two graphs with a particular structural form during the first day. The effect of prior structural knowledge was then tested on the following day by examining participants' learning of a third graph (Fig. 1). To test for transfer of structural knowledge, in each experiment, we divided participants into two different groups. One group was exposed to graphs with the same structural form (but different pictures) on both days. The second group was exposed to graphs with different structural forms (and pictures) on each day (Fig. 1a). This design allows us to control for all effects that are independent of the structure of the graphs; since the task is independent of the structural form, and its identity is not explicitly observed by the participants.

**Associative representation.** Learning such graphs and creating a cognitive map can be accomplished using different types of representations. One solution to such a problem is a conjunctive representation of the stimuli and their relationships; the

relationships between the stimuli are encoded by the associations between the representations of the stimuli themselves (Fig. 2). An example of such representation is the SR[3]. Here, the representation of each state (in our setup each stimulus defines a state) encodes the probability to reach any other states in the future. Using such a representation it is possible to determine whether two stimuli are neighbouring nodes on a graph and even to navigate on a graph.

**Inferring and transferring graph structure.** A different option for learning a cognitive map is to represent the structure of the graph and its stimuli using different, disentangled representations (Fig. 2). We considered a HMM. The Markov assumption is that each latent state depends directly only on the state at the previous time step. In other words, the past is independent of the future conditioned on the present. This dependency is captured by the transition matrix, $A$. Each entry, $A_{ij}$, in this matrix represents the probability to move from state $i$ to state $j$. A second matrix, the emission matrix $B_{ik}$, represents the probability that state $i$ will emit observation (particular stimulus) $k$. Together, both these matrices describe the probability of a sequence of observations. The HMM framework is promising because it maintains separate representations of transitions and emissions and therefore can easily generalise transition structures to new sensory stimuli (Fig. 2). However, we consider two extensions to vanilla HMMs.

First, because transition matrices ($A$) of the same structural form may not be identical (e.g. different number of nodes), we need a flexible representation of transition structure. We

approximate the transition structure using basis sets for structural knowledge (Fig. 3b, c, d, see below and 'Methods' for further details). Second, we propose a method for inferring, amongst candidate basis sets, the one that best fits the current task. We assume that each common structural form is represented by a basis set and this basis set is known to the agent. The agent needs to infer the underlying structural form and exploit the knowledge of the basis set to estimate the transition matrix of the current task. Once the structural form is inferred, the size of the graph should also be inferred. The agent should then adjust the basis set according to the

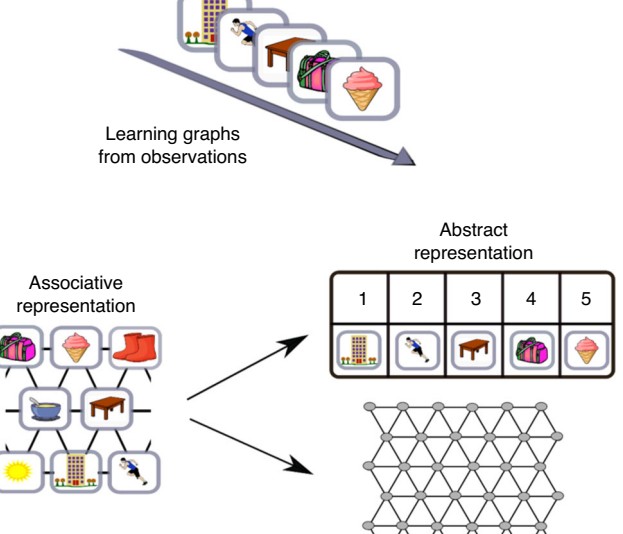

**Fig. 2 Associative and abstract representation of transition structure.** Learning underlying graph structure from observations of pictures. Graph can be represented by learning the associations between the stimuli (associative representation). Such representation is conjunctive, as the relations between the stimuli are encoded using the associations between the representation of the stimuli themselves. This type of representation does not allow generalization and knowledge transfer. Representing the graph using two separate matrices, the transition and emission matrices allows generalization over graph structure.

inferred graph size and estimate the transition structure of the current task (Fig. 3d, see 'Methods' for details).

The problem at hand can be formulated as a hierarchical generative model of graphs[25]. Each structural form, using the basis set representation, can generate a particular transition structure according to a vector of parameters ($\theta$) that defines the particularities of the current graph, such as size. Together with the emission matrix (B), the observations (O) can be generated (Fig. 3a). The task for the agent is to infer the current structural form ($S_f$), graph size and the emission matrix. The agent first uses approximate Bayes (see 'Methods') to infer the structural form and graph size:

$$p(S_f, \theta | \vec{\mathbf{O}}) \propto p(\vec{\mathbf{O}} | \theta, S_f) p(\theta | S_f) p(S_f) \tag{1}$$

Where:

$$p(\vec{\mathbf{O}} | \theta, S_f) = \int p(\vec{\mathbf{O}} | B, \theta, S_f) p(B | \theta, S_f) dB. \tag{2}$$

While doing so, the agent calculates the maximum likelihood estimates of the emission matrices for each considered graph (see 'Methods'). The emission matrix that corresponds to the inferred structural form and graph size is then chosen to represent the current task. Using this method, the agent was able to infer the correct structural form (Supplementary Figs. 2 and 3) and graph size (Fig. 6a and Supplementary Fig. 1). Following the estimation of transition and emission matrices, the agent can estimate the distances (number of links) between observations (see 'Methods'). Indeed, when asking the agent to report which of two pictures is closer to the target picture, similarly to participants, the agent was able to perform it correctly (Supplementary Fig. 1).

**Basis sets definition**. The problem at hand is representing abstract structural knowledge. It was previously shown that topology of a graph is well represented by eigenvectors of the graph Laplacian[20]. The graphs that we considered here are symmetrical; therefore, the eigenvectors of the graph Laplacian and the transition structure are the same. For basis representations of the hexagonal structures, we therefore, chose eigenvectors of a hexagonal graph transition matrix (see 'Methods', Fig. 3b). These eigenvectors have previously been shown to resemble Entorhinal cortex grid cells[2,26]. Because the community-structure graph is not translationally invariant, it can be more compactly

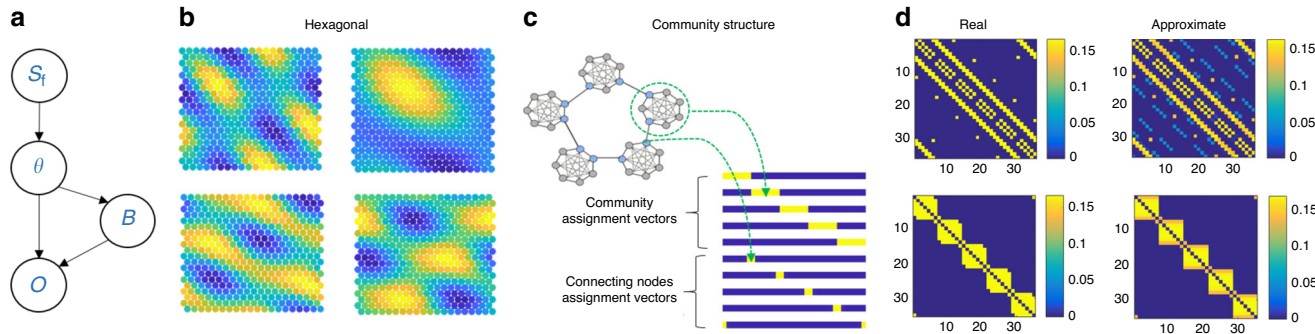

**Fig. 3 Inferring graph structure rather than learning it using a basis sets representation for structural knowledge. a** We present a generative model for graphs. Each graph belongs to a structural form ($S_f$). Given a structural form, graph size ($\theta$) is sampled from prior distribution ($p(\theta | S_f)$) and the transition matrix is approximated. Given a transition matrix ($A_{sf}^{\theta}$, that is determined by the form and the dimensions) an emission matrix (B) is sampled. From these two matrices, the sequence of observations (O) can be generated. **b** Basis sets for Hexagonal grid, few examples. **c** Basis sets for a community structure. Basis sets can allow direct inference of important graph states without the need of further computation. In a graph with underlying community structure, the connecting nodes (blue circles) are important; knowing them allows fast transitions between communities. A basis set that contains explicit connecting nodes assignment vectors allows the direct inference of their identity by learning the emission matrix. **d** The transition matrices can be approximated using Basis sets for structural knowledge. Upper panels: correct and approximated transition structure for Hexagonal grids with 36 nodes. Lower panels: real and approximated transition matrices for a graph with underlying community structure (35 nodes).

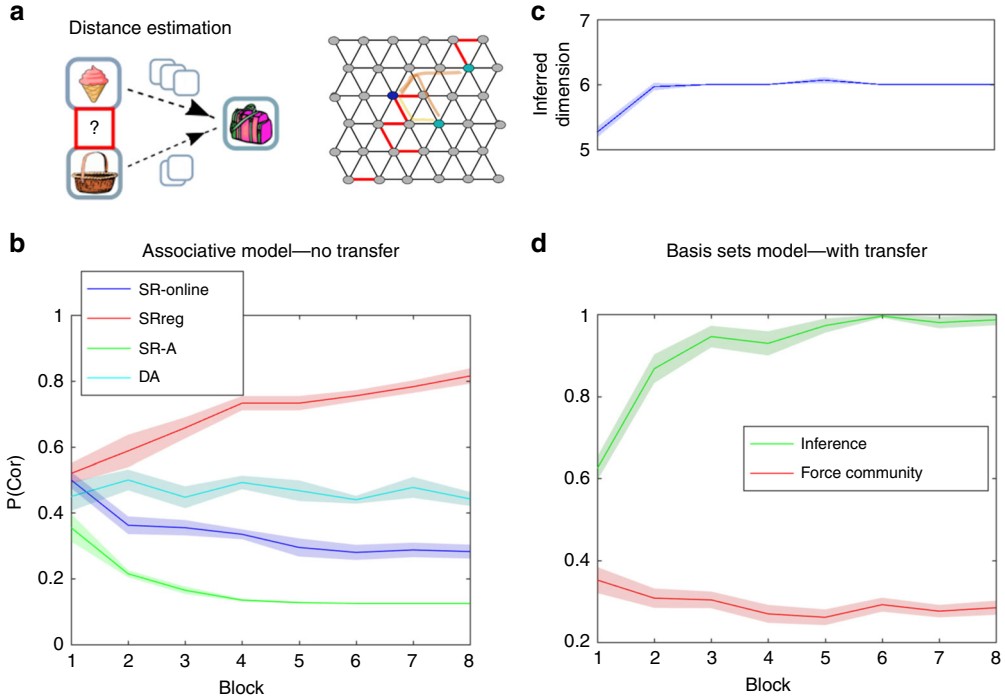

**Fig. 4 Inference of unobserved links (Hexagonal graph). a** Inferring the existence of unobserved edges (links). Left—the task: The agents had to indicate which of two nodes (pictures) has smaller number of links to the target. With only observed links, the number of links to the target was identical. Right—red edges indicates missing links on the graph. For example, the two nodes that are marked with light blue have the same number of observed links to the target node (marked with dark blue circle), while the number of links that connect these two nodes to the target is different on the complete graph. **b** When learning from pairs that were sampled randomly (not in succession) while some of the links (pairs) were never observed, simple associative models as learning transition matrix (DA) or simple SR (SR-online: learning using TD-SR[44], SR-A: calculating SR from the learnt transition matrix) could not infer the existence of the unobserved links and solve the task (it in fact solves it worse than chance). Agents that use a filtered SR representation (SRreg) could answer these questions better than chance. Shadows are the standard errors of the mean (SEM), the centre is the mean. **c** When learning from pairs that were sampled randomly (not in succession) while some of the links (pairs) were never observed), the basis set agent, that transfers abstract structural knowledge, was able to infer the structural form (Supplementary Fig. 2) and graph size correctly. **d** Further, the agent was able to infer the existence of links that were never observed and determined correctly, which of two pictures is closer to a target picture, according to the complete graph (green). The agent could do so even though the number of observed links between the two pictures and the target was identical (p(cor) corresponds to the average fraction of correct answers out of 40 questions in each block). When the agent was forced to infer a community structure (red), it answered these questions worse than chance. Shadows are the standard errors of the mean, the centre is the mean.

represented using bases that take two distinct forms—a set of bases for community membership and a set for connecting nodes, where 'connecting nodes' refers to nodes that connect two communities (Fig. 3c). As we will show, this has the additional benefit of rapidly inferring connecting nodes, which leads to behavioural advantages. This basis set resembles the transition structure eigenvectors of such graphs but allows higher flexibility in representing communities that lie on different, higher level, structures (such as rings, grids etc., see Supplementary Notes 1 & 3 for further discussion).

**Inferring unobserved trajectories**. Determining whether two pictures are neighbours on a graph, or navigating on a graph, can be achieved by both types of models—associative learning and HMM—as these tasks require knowledge of the associations between the pictures. However, the aim of this work is to examine the ability of models and humans to represent and transfer abstract structural knowledge. Here, we show that abstract structure of the task can be exploited for inference of unobserved edges and this abstract structural knowledge can be transferred between graphs.

To test inference of unobserved edges, we tested the models (and later the human participants) on a more difficult problem. Here, instead of a random walk, the models learn the graph by

pseudo-random sampling of pairs of adjacent states. This is a harder problem than inferring the graph from random walks, as loop-closures (when the participant sees an entire trajectory, starting from a certain state and returning to the same state) are far less frequent so the number of possible graphs consistent with the observations remains high for longer. However, this allowed us to perform a key manipulation. We could selectively omit key edges in the graph (red edges Fig. 4a) without changing the local sequence statistics (because there were no sequences, only pseudo-random presentations of adjacent pairs). We could then ask if the agents (and later humans) could infer the existence of these omitted edges. To test whether the models (and later humans) could infer unobserved links, we asked questions of the following form: Which of two observations is closer to the target state? In each case, the two observations were the same distance to the target, given the observed edges, but one of the observations would be closer if the model (or human) had inferred the existence of the omitted edges.

Simple associative models, such as learning the transition matrix between the pictures themselves or learning SR, cannot solve such a problem (Fig. 4b). Following Stachenfeld et al.[2], we spectrally filtered the SR that is currently being learnt using its own eigendecomposition; we reconstructed the SR using the seven most informative eigenvectors only. Such filtration smooths

over the unobserved edges, which then allows the agent to answer such questions better than chance without the need of knowledge transfer (Fig. 4b, $p < 0.001$ one-tailed $t$ test). Our basis sets model was able to infer the correct structural form (Supplementary Fig. 1) and graph size (Fig. 4c), it exploits this knowledge to infer the unobserved edges better than chance and answer the questions above correctly (Fig. 4d, $p < 0.001$ one-tailed $t$ test, see also Supplementary Fig. 1). Hence, we can conclude that basis sets, as a compressed representation of transition structures, allow estimation and inference of the currently relevant transition structure and therefore enable the prediction of edges that were never observed.

We then wanted to check whether humans solve such problems using transfer of abstract structural knowledge or whether they exploit a smoothed associative representation. If participants can solve the task without the need of prior knowledge it will imply that associative learning is enough, while if their performance depends on prior knowledge, we can conclude that humans do represent and transfer abstract structural knowledge. To test whether humans infer the existence of unobserved edges by using clever smoothing of noisy representation or whether an abstract knowledge is being transferred, we designed the following two experiments.

**Humans infer unobserved trajectories**. Do humans use prior structural knowledge of the underlying graph structure to infer the existence of transitions that were never observed? We performed graph-learning experiments where participants learned three large graphs (36 nodes with degree of 6, Fig. 1a). We tested whether participants can infer (or learn) the underlying graph structure and apply this knowledge to a new graph with new stimuli. Participants were segregated into two groups. They performed the task on 2 successive days (Fig. 1a). During the first day, one group learned two graphs with an underlying hexagonal structure while the second group learned two graphs with an underlying community structure. On that day, the graphs were learnt by observing a sequence of pictures that are taken from a random walk on the graphs.

Following the analogical reasoning theory in psychology[6], we hypothesised that the experience during the first day shaped the prior expectations over the underlying structural forms during the second day, as participants associated the experienced graph statistics with our task. Participants who learned hexagonal graphs during the first day should expect a hexagonal graph on the second day, while participants who previously learned graphs with underlying community structure should expect to learn again a graph with a community structure. We therefore asked whether participants can infer the underlying structural form during the first day and then use it as prior knowledge during the second day. Notably, if they do, they will be able to infer the existence of transitions they have never observed (as in the model). Therefore, as with the model described above, both groups of participants learned hexagonal graph on the second day by observing pairs of adjacent pictures. As with the model, pairs were sampled pseudo-randomly (i.e. neighbouring pairs were not sampled in succession) and many pairs were omitted. That is, many transitions were never explicitly observed by the participants (depicted in Fig. 4a red lines). We aimed to test whether participants could use structural knowledge from the first day to infer the existence of these unobserved transitions.

We used the exact same testing procedures as with the models above to examine participants' ability to infer the existence of a link that was never observed explicitly; participants had to indicate which of two pictures is closer to a target picture; no feedback was given for this type of questions (more than 200 questions for each participant). As with the models, the two

pictures were the same distance to the target, given the observed links, but one was closer if the existence of the missing link was inferred. Only participants who were able to complete 'missing links' using knowledge of the underlying graph structure could answer these questions correctly. Indeed, participants who had experienced the hexagonal structure on different graphs the previous day, performed significantly better than control participants who had experienced graphs with underlying community structure (Fig. 5, left: all questions, right: 'missing links' questions only. $<Phex(cor)> = 0.54$, $<Pcl(cor)> = 0.5$, $t = 2.29$, $p$-value $= 0.016$ for inference questions, $<Phex(cor)> = 0.56$, $<Pcl(cor)> = 0.52$, $t = 2.54$, $p$-value $= 0.0068$ for all questions, df $= 58$, one-tailed $t$ test, the results are significant for two-tailed test as well). These results indicate that, similarly to our basis sets model, participants extract sophisticated structural knowledge of the problem that generalises across different sensory realisations. They were able to transfer knowledge from one day to the other and use this knowledge to guide their decisions and infer unobserved trajectories. This effect cannot be driven by non-inferential approximations of graph distances, such as the smoothed SR model[2,3], as all such measures consider each graph independently and are therefore invariant to the structural form of the previous day's graph (with different stimuli).

Notably, this effect is driven by a subset of participants (Fig. 5). This subset performs the inference extremely reliably (individual $p$-values $< 10^{-5}$). We conclude that despite the group wise significant effect, only a subset of participants were able to exploit the transferred structural knowledge of the hexagonal graph. This may be because of the difficult nature of the graph tasks, with many states, no visual or border cues to help define the

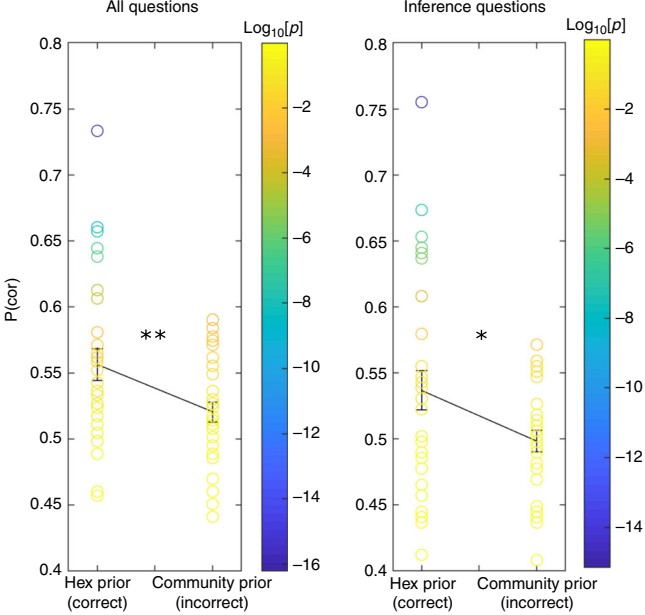

**Fig. 5 Transfer of structural knowledge allows inference of unobserved links (Hexagonal graph).** Participants had to indicate which of two pictures is closer to a target picture. Participants that reached the second day of our task with the correct prior expectation over the structural forms performed significantly better in such task compared to participants with the wrong structural prior (left panel) (30 participants in each group). They were able to answer these questions significantly above chance even when there were links that were never observed, and they had to choose between two pictures with an identical number of observed links to the target (right panel). One-tailed $t$ test. **$p < 0.01$, *$p < 0.05$. Error bar: SEM. Colorcode: $Log_{10}(p$-value).

graph, and no feedback. Nevertheless, participants who did infer the structure could use it to perform inference at levels far above chance. We should further note that our models have a perfect memory and no implicit noise, therefore, we expect them to be better than human participants.

**Using structural knowledge to set advantageous policies.** Not only can structural knowledge be used to infer unobserved transitions, it can also be used to direct advantageous policies. For example, while navigating on a graph with a community structure, agents with no structural knowledge will spend large periods trapped in a single community. A simple policy of 'prefer connecting nodes' overcomes this problem. When the correct underlying structure is of community structure, our basis sets model can infer it correctly (see Supplementary Fig. 2). Our model also infers the number of communities correctly (Fig. 6a upper panel). Using this particular basis set for transition matrix estimation allows direct identification of connecting nodes (Fig. 3c). Indeed, the identity of the connecting nodes is recovered correctly during the learning of the emission matrix (Fig. 6a low panel, see 'Methods').

To establish whether participants can infer the existence of community structure and use a prior over the structural forms to inform their behaviour, we constructed a second experiment. In this experiment, participants were also segregated into two groups. As before, one group learned two hexagonal graphs and the other group learned two graphs with community structure during the first day. However now, both groups learned from random walk and navigate on a community-structured graph during the second day (Fig. 1a). Participants who learned graphs

with underlying community structure on the first day indeed performed better on the second-day navigation task (number of steps to the target is shorter, Fig. 6b upper panel, $D_{t=0}$, is the initial distance between the starting picture and the target, see 'Methods' for complete statistical values). Furthermore, they learned the associations faster. While learning the associations, participants determined their own learning pace by choosing when to observe the next pictures pair. Participants who expected a graph with underlying community structure spent less time on learning each pair of pictures than participants who expected a Hexagonal graph (Fig. 6b upper left panel, $p = 0.003$, $t = 3.19$, df = 38, two-tailed $t$ test). These results suggest that participants' behaviour was affected by structural prior; the previously experienced graph structure affected the learning policy of participants. The learning policy that is adjusted to underlying community structure leads to faster learning and better task performance. One likely possibility is that, instead of learning the individual pairwise associations, participants simply inferred the community structure and assigned each node to the current community, while identifying the connecting nodes.

In order to understand how different behavioural policies lead to different performance, we examined participants' choices during navigation. During the navigation part of the task, participants had to choose between two pictures (to get closer to the target) or skip and sample a new pair (if they thought both pictures took them further away). We examined participants' choices during all trials in which one picture was a connecting node and the other was not. Participants who had the correct prior chose connecting nodes significantly more than participants who had the wrong prior ($p = 0.03$, $t = 2.25$, df = 38 two-tailed

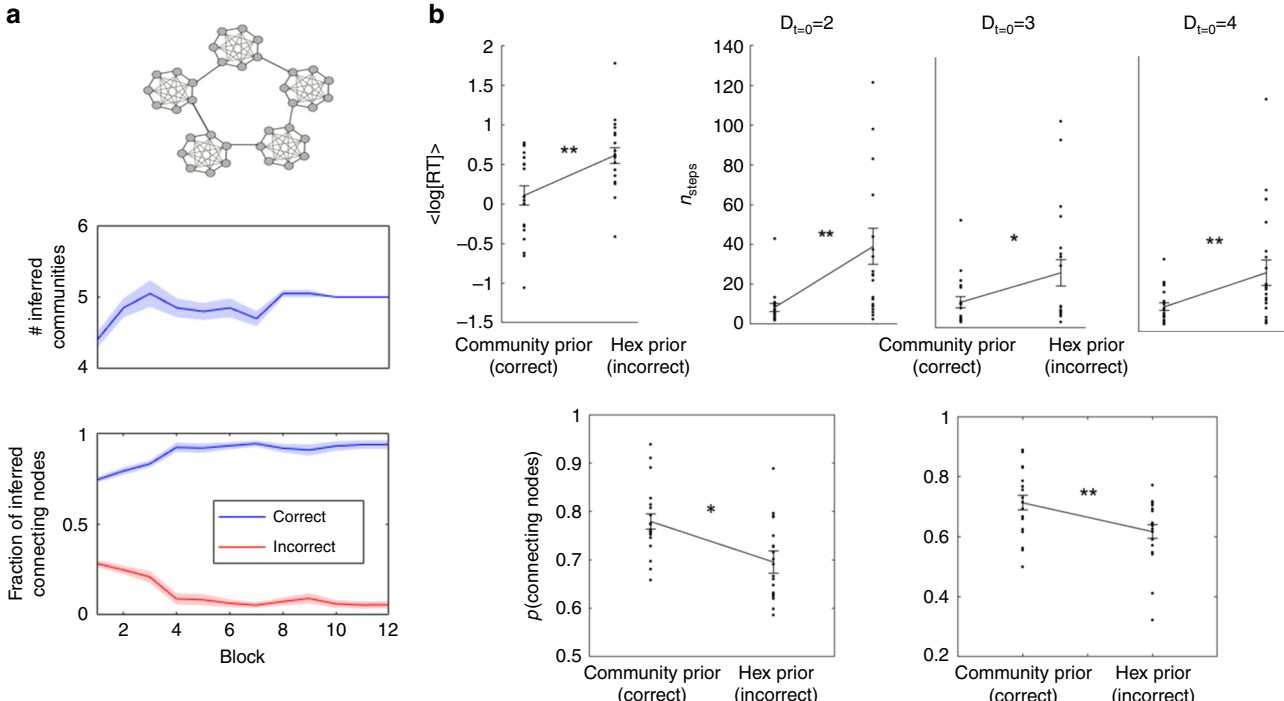

**Fig. 6 Policy transfer: learning graphs with underlying community structure. a** Our agent was able to infer the correct number of communities (middle panel, averaged inferred number of communities over 20 simulations). It was also able to infer the identities of the connecting nodes (lower panel, inferred number of nodes divided to the number of connecting nodes according to the inferred graph size, see 'Methods'). Shadows are the SEM, the centre is the mean. **b** Participants with correct structural prior spend less time on learning the associations between the pictures (RT = response time for changing to the next pair, upper panel—left). The number of steps to the target ($n_{steps}$) is significantly lower for participants with the correct structural prior (upper panels, $D_{t=0}$ is the initial number of links between the current picture and the target). During navigation, participants with the correct prior over the structural forms choose connecting nodes more frequently (lower panel—left), they do so even if this choice takes them far away from the target (lower panel—right). Error bars are the SEM, the centre is the mean. *$p < 0.05$, **$p < 0.01$ (20 participants in each group).

*t* test, Fig. 6b left lower panel). Furthermore, this cannot be driven by better inference of graph distances as they chose connecting nodes more frequently, even if this choice was the wrong choice ($p = 0.006$, $t = 2.88$, df $= 38$, two-tailed *t* test, it took them far away from their target, Fig. 6b right lower panel). These results imply that participants with the correct structural prior behave according to a policy of 'prefer connecting nodes'.

Here, inference of structural knowledge can lead to the transfer of two different types of knowledge. First, the transfer of the abstract transition structure (as participants did in the previous task). Second, the transfer of unique behavioural policy that is tailored to that particular structural form. We cannot identify here whether our results originated from policy transfer only, or whether participants also transfer abstract knowledge of the transition structure itself. According to our model, inference of structural knowledge allows transfer of the relevant basis set. Basis set for community structure enables inference of connecting nodes' identity immediately when learning the emission matrix. Therefore, exploiting this basis during the task should enable faster identification of these nodes. This implies that transfer of structural knowledge itself (at the form of the relevant basis set) will lead to better identification of these special nodes. Further, basis set representation generalizes over all tasks that are governed by the same structural form, therefore, they allow the semantic understanding of connecting nodes and the generalization of such policy. This experiment supports this idea but other types of representation for connecting node identity may enable the same behaviour.

Being trapped in a community for a long period can be frustrating. Hence, escaping from a community might be perceived as rewarding. One possible explanation for the choice of connecting nodes is therefore simply that they have been more rewarding in the past. However, this cannot explain the effect we observed. We compared the behaviour of two groups of participants that did the exact same task and only differ by the underlying graph structures that have been learned the day before. Participants with the wrong prior spent longer within communities and would, under this argument, experience greater reward when escaping them. Therefore, the value of connecting nodes assigned by the participants with the wrong prior should be higher. If participants chose according to value differences, we should see the opposite effect. This model-free effect therefore runs opposite to the behaviour we observe.

## Discussion
We have shown, using a graph-learning task, that participants are able to transfer abstract structural knowledge. They were able to transfer abstract transition structure of the task and structurally relevant behavioural policy. They exploited the transferred structural knowledge to infer the existence of unobserved trajectories, identify important task states and improve performance on the task. Using a computational model, we have suggested a representation for structural forms that allows generalization over particularities of the current task and enables transfer of abstract structural knowledge. Each structural form is represented by a particular basis set that encodes flexibly the transition structures that belong to that form. Each basis vector can be stretched and compressed according to the inferred graph size, hence, the set enables compressed and generalizable representation of the transition structure[13]. We demonstrated that having such representations enables the correct inference of the structural form governing the associations between states in the task. Approximating the current transition matrix using a basis set allows inference of routes that have not been taken before and inference of the identities of important task states. Our current

experimental results suggest that humans do exploit abstract structural knowledge, but whether they achieve this via basis representations requires further experiments.

We compared our model to models of associative learning. These models represent relations between states of the task using associations between the representations of the states themselves. Although simple associative models cannot infer the existence of unobserved associations, smoothing of such representations allows these inferences. However, our behavioural experiment suggests that humans do exploit knowledge transfer for such inferences. The ability to transfer structural knowledge requires that participants infer or acquire the correct representation of the hidden structure of the graph during the first day. One possibility is that structural knowledge is acquired slowly via experiencing different scenarios that share the same underlying structural form during our life[27–29]. In the context of our task, we suggest that the experience during the first day shapes the prior over structural forms on the second day[6]. Then, participants with the correct structural prior are able to infer the correct structural form faster, estimate better the current transition matrix and transfer the correct behavioural policy, therefore achieving better performance in the task. We would like to note that it is possible that the brain exploits both abstract representation and associative learning strategy that interact and complement each other.

In the current work, we considered two types of structural forms to introduce the idea of basis sets representations for structural knowledge. Theoretically, the idea of basis sets representation for structural knowledge can be extended to other structures that are common in nature, such as rings and hierarchies. For example, eigenvectors of transition matrices with underlying structural form of hierarchy contain the information on the layer of nodes in the hierarchy (see supplementary Note 4). Similarly, to connecting nodes in a graph with a community structure, this information can be beneficial in performing a task and allows the learning of the meaning of each layer over a variety of tasks with the same underlying structural form.

Generalizing structural knowledge in the form of structural forms have been suggested previously by Kemp et al.[9] They have suggested a generative model for constructing a graph using general structural elements, each belonging to a different structural form[9]. As this model exploits abstract structural knowledge, we expect it to perform well on our tasks as well. Our choice of basis sets for representing structural knowledge is inspired both by spectral graph theory[13,20,30] and mainly by existing research on the hippocampal–entorhinal system. Entorhinal cortex consists of cells that have hexagonal activity patterns while the animal walks freely in an arena (grid cells)[31]. This activity pattern resembles the patterns of transition matrix eigenvectors of hexagonal graphs[32,33]. Further, grid-like representations emerge in environments/tasks that are not spatial but share similar statistical structure[34–36]. These observations may suggest that grid cells can be used as basis functions for all environments in which the associations between the states are governed by the rules of 2D Euclidean space. Further, basis sets representation for structural knowledge is beneficial as it allows direct inference of the structure of the task without the need of mental simulations.

It has been suggested that grid cells are created using attractor neural networks[37,38]. This suggestion is supported by the observation that their activity correlation pattern is maintained during sleep[39,40]. Further, grid cells remap (their activity pattern is shifted) in new environments but their hexagonal activity pattern remains[41]. These observations may suggest that grid cells activity pattern is stably represented and therefore can be recalled in new environments that share the same underlying statistical pattern of transition structure, in accordance with our basis sets hypothesis. Further, our hexagonal graphs were a torus, such that there were

no boundaries. Introducing boundaries and a policy in which the agent prefers to stay near the boundaries, similarly to animals behaviour, creates an asymmetrical transition structure. The right eigenvectors of such representation hold the variation in transitions and have patterns that resemble EC boundary cells[42] (in addition to hexagonal patterns, see Supplementary Note 2). We can think of a boundary cell as part of a basis that captures special nodes in translational invariants graphs.

Inspired by graph theory, reinforcement learning and the activity patterns in the hippocampal formation, we suggest that the brain may represent structural forms in a form of basis sets. Using modelling, we show that such basis sets allow transfer of structural knowledge that is relevant to the current task. Our behavioural experiments demonstrate that humans can transfer abstract structural knowledge and exploit it in a new task.

## Methods

**The generative model (basis sets model).** In this work, we follow Tenenbaum et al.[18] in suggesting that humans represent structural knowledge as structural forms. Each structural form is a family of graphs in which the nodes of the graph are organized according to a particular rule. For example, hexagonal grid connectivity patterns will always present a translational and rotational symmetry, or in community structure, the nodes within a community will be highly interconnected, while the connectivity between communities is sparse. We assume that on each new task, humans infer the structural form that best fits the graph of the current task. Following this inference, they can transfer the relevant information that represents this structural form.

Following Kemp and Tenenbaum[9], we formalized the inference over structural forms using a hierarchical generative model of graphs. In our task, the observations of the agents and participants are Markovian (each state depends only on the state before) and follow a transition matrix that is characterized by an underlying graph structure. Each graph belongs to one of the structural forms that are considered in our experiments (hexagonal grid and community structure). We assume that each task structural form ($S_f$) is generated by sampling from a uniform distribution over the structural forms. Given a structural form, the graph dimensions ($\theta$) are sampled from a prior distribution that is unique for each structural form (see below). Together, $S_f$ and $\theta$ fully determine the transition matrix of the graph ($A_{sf}^{\theta}$). Then, given a transition matrix, the emission matrix ($B$) is sampled (in the following, we will not find the posterior of $B$, therefore we do not state any prior for $B$ here). Using these two matrices, the observation (**O**) can be generated.

$$S_f \sim p(S_f) = Cat\left(\frac{1}{2}1\right) \tag{3}$$

$$\theta \sim p(\theta|S_f) = Cat\left(\frac{1}{3}1\right) \tag{4}$$

Where *cat* is the categorical distribution.

**Model inversion (basis sets model).** The task of the agent is to infer the hidden states of this generative model; given a set of observations, the agent should infer, using Bayes rule, the structural form and graph dimensions ($\theta$) that characterized the graph of the current task or environment.

$$p(S_f, \theta|\vec{\mathbf{O}}) \propto p(\vec{\mathbf{O}}|\theta, S_f)p(\theta|S_f)p(S_f) \tag{5}$$

Where:

$$p(\vec{\mathbf{O}}|\theta, S_f) = \int p(\vec{\mathbf{O}}|B, \theta, S_f)p(B|\theta, S_f)dB. \tag{6}$$

Here the integral is over all possible values of the entries in the emission matrix $B$. Solving this integral is hard, therefore, we have approximated it by using the Bayesian Information Criterion (BIC):

$$p(\vec{\mathbf{O}}|\theta, S_f) \sim e^{\frac{-BIC}{2}} = e^{\log \hat{L} - \frac{N}{2} \cdot \log(k)} \tag{7}$$

Where $N$ is the number of states in the graph, $k$ is the number of observations and $\hat{L} = p(\vec{\mathbf{O}}|\theta, S_f, \hat{B})$ is the likelihood of the sequence of observations with $\hat{B}$ as the maximum likelihood estimate of the emission matrix. The transition matrix ($A$) is fully defined by the structural form and the dimension of the graph. As the observations depend only on the transition and emission matrices, we can write the likelihood as: $\hat{L} = p(\vec{\mathbf{O}}|\theta, S_f, \hat{B}) = p(\vec{\mathbf{O}}|\hat{A}_{S_f}^{\theta}, \hat{B})$ where $\hat{A}_{S_f}^{\theta}$ is an estimated transition matrix (see below). Our model is an HMM, therefore, we can exploit a variant of the Baum–Welch algorithm[43] to estimate $B$ from the observations and calculate the likelihood $\hat{L}$ (see Supplementary Methods for details). The Baum–Welch algorithm gives a maximum likelihood estimate for the transition and emission matrices as

well as the likelihood itself. Here, instead of learning the transition and emission matrices from the data (**O**), we learned only the emission matrix and assumed that the transition matrix is known; for each structural form and graph size that were considered, we approximated the transition matrix using the relevant basis set (see below). For each approximated transition matrix we estimated $\hat{B}$ and $\hat{L}$. Using these quantities, we estimated $p(S_f, \theta|\vec{\mathbf{O}})$ and inferred the current structural form and dimension.

The structural form of the current task is inferred by calculating the posterior and choosing its maximum (MAP):

$$p(S_f|\vec{\mathbf{O}}) = \sum_{\theta} p(S_f, \theta|\vec{\mathbf{O}}) \tag{8}$$

Following the inference of the structural form the current graph size is inferred using MAP of:

$$p(\theta|\vec{\mathbf{O}}, S_f) \propto p(\vec{\mathbf{O}}|\theta, S_f)p(\theta|S_f) \tag{9}$$

**Approximating the transition matrices using basis sets.** To allow generalization over particularities of the current graph structure such as its dimensions ($\theta$), the transition matrices are approximated using basis sets ($U_{Sf}$) for structural knowledge. Each structural form is represented by a unique basis set and transition matrices of all graphs that share structural form are approximated using this set (see below the definitions of the basis set for each of the structural forms). Instead of learning the transition and emission matrix from the data ($\vec{O}$), we infer the transition matrix under the assumption that the task transition matrix can be approximated by those basis sets that are already known. Therefore, once the agent solved the inference problem over structural forms and graph size, it can use its prior representations of possible basis sets to estimate the new transition matrix.

For each structural form, given a particular graph size, the basis vectors in the set can be stretched and compressed, using interpolation to adjust for the currently estimated graph (matlab imresize). The approximated transition matrix becomes: $A_{sf}^{\theta} = f(U_{sf}^{\theta} \cdot S_{sf} \cdot U_{sf}^{\theta T})$, where $U_{sf}^{\theta}$ is the adjusted basis set, $S_{sf}$ is a diagonal matrix of weights (eigenvalues in the hexagonal grid graphs and ones in the community-structure graphs), and $f$ is a threshold linear function. We then subtract the diagonal and normalize the matrix (see Fig. 3d).

**Inferring graph size.** We assumed, for simplicity, that there are three different possible graph sizes for each structural form, therefore the prior probabilities of $\theta$ are uniform within this set and zero otherwise. We further assumed, for simplicity, that the two dimensions of the hexagonal grid are equal and the number of nodes in a community is also equal, hence $\theta$ defines a vector of possible number of nodes in a graph. For hexagonal graphs we considered $N = [25,36,49]$, for a graph with underlying community structure $N = [28,35,42]$ with equal prior probability. We emphasise that the basis set ($U_{sf}^{\theta}$) for each transition structure within a structural form is a scaled or truncated version of a general basis set for that structural form.

**Estimating distances between two pictures.** Using the inferred graph transition and emission matrices, the agent approximated the distance between two observations. As the transition structure is approximately known, we estimated the distance matrix between two abstract states ($z_i$) on the graph using this approximation; we adopted a threshold function of the transition matrix to estimate the Adjacency matrix and then estimated the abstract distance matrix ($D(z_m, z_k)$) using it. As the distance matrix represents the distances between abstract states, the distances or the number of steps between the observations themselves is calculated by:

$$\tilde{D}(O_i, O_j) = \sum_{k,m} p(z_m|O_i)D(z_m, z_k)p(z_k|O_j) \tag{10}$$

The emission matrix gives us $B_{z_m}(O_i) = p(z_m|O_i)$, as $p(z_m)$ is uniform, we inverted $p(z_m)$ by:

$$p(z_m|O_i) = \frac{p(O_i|z_m)}{\sum_n p(O_i|z_n)} \equiv \tilde{B}_{mi} \tag{11}$$

Then, we multiply the abstract distance matrix by $\tilde{B}$ to get the distance matrix between the observations:

$$\tilde{D} = \tilde{B} \cdot D \cdot \tilde{B}^T \tag{12}$$

Using this matrix, we calculated the decision of the agent when selecting between presented pictures (that is, observations; each observation corresponds to one state on the graph), when tasked with selecting the picture (state) which is closer to the target picture (state) in the graph. We would like to note here that the actual emission matrix in our task is not probabilistic and it is an identity matrix. When the agent estimates the emission matrix from the observations it converges to any permutation matrix that maintains the symmetry of the graph. There are other approximations for the distances between observations that can be adopted which take into account the probability structure of the transition matrix, such as the SR[3].

**Connecting node inference**. For estimating the number of stimuli (that is, observations) that are correctly inferred to be connecting nodes, we calculated the fraction of correctly inferred connecting nodes as: $fc_b = \frac{Is_c \cdot B_c \cdot Ip_c^T}{nIc}$ for each simulated block. Here, $Is_c$ is a vector of the size of the inferred number of states (that is, graph size), where entries corresponding to connecting nodes are equal to one, with the remaining nodes equal to zero. Similarly, $Ip_c$ is a vector of size equal to the number of stimuli (observations), where values are equal to one when that stimulus corresponds to a connecting node, and zero otherwise. $nIc$ corresponds to the number of connecting nodes in the inferred graph. The fraction of nodes incorrectly inferred as connecting nodes is defined as: $fIc_b = \frac{Is_{nc} \cdot B_c \cdot Ip_c^T}{nIc}$ where $Is_{nc} = 1 - Is_c$.

**Basis sets**. *Community structure*: The number of nodes within each community is considered constant for simplicity. There is an assignment vector for each community with a value of one for each node that belongs to that community and zero otherwise. The number of such vectors is determined by the number of communities that are currently considered. Further, there are 'connecting node assignment vectors' for each community, which give a probability for a node in a certain community to connect to another node in another community, the probability for a second connecting node is lower. This probability is a Gaussian with a number of connecting nodes as its variable. See supplementary Note 3 for further discussion on this choice.

*Hex*: The eigenvectors of a large hexagonal graph were computed. We kept as a basis set the 12 most informative eigenvectors (excluding the constant). We then resize the eigenvectors according to the size of the graph that is currently considered using standard interpolation method (matlab imresize).

**Successor representation model**. The successor representation is defined as:

$$\mathrm{SR} = \sum_t \gamma^t A^t = inv(I - \gamma A), \qquad (13)$$

where $A$ is the transition matrix, $I$ is the identity matrix and $\gamma$ is a discount factor.

The SR can be updated within blocks, after updating the transition matrix online, or using TD learning[44]. After observing a transition at time step $t+1$ of $s_t \to s_{t+1}$ the SR is updated according to:

$$\mathrm{SR}_{t+1}(s_{t+1}, s') = \mathrm{SR}_t(s_t, s') + \alpha[I(s_t = s') + \gamma \mathrm{SR}_t(s_{t+1}, s') - \mathrm{SR}_t(s_t, s')], \quad (14)$$

where $\alpha$ is the learning rate. We made the SR symmetrical at the end of each block.

Spectral regularization (filtering the SR using its own eigendecomposition):

We have calculated the eigendecomposition (using SVD) of the SR that has been learnt using TD. We then calculated the regularized SR by: $\mathrm{SR}_s = U_m S_m U_m^T$, where $U_m$ is the matrix of the $m = 7$ most informative eigenvectors of the symmetrised SR and $S_m$ is a diagonal matrix with the $m = 7$ largest eigenvalues on its diagonal. $\gamma = 0.8$. The number of simulations for Fig. 4b is 10.

**Behavioural experiments**. *Participants*: We recruited 100 participants, 60 participants for experiment 1 (30 in each group) and 40 participants for experiment 2 (20 in each group). All participants are UCL students with an average age of 23.5.

The study was approved by the University College London Research Ethics Committee (Project ID 11235/001). Participants gave written informed consent before the experiment.

**Graphs structure**. Experiment 1: transfer of hexagonal structure: Each hexagonal graph consisted of 36 nodes and periodic boundary conditions as shown in Fig. 1.

Experiment 2: transfer of community structure: Each graph consisted of five communities with seven nodes each. Within a community, each node was connected to all other nodes except for connecting nodes that were not connected to each other but were each connected to a connecting node of a neighbouring community (Fig. 1). Therefore, all nodes had a degree of six, similarly to hexagonal graphs. Our community-structure graph had a hierarchical structure, wherein communities are organised on a ring. We hypothesized that inference of the second order structure of a ring and transfer of this structure from day one to day two will allow participants to infer a missing link that closes the ring. We therefore introduced a missing link during the second day (see Supplementary Note 1 for the results).

**Experimental procedures**. Participants learned two graphs with the same underlying structure but different stimuli during the first day. Stimuli were selected randomly, for each participant, from a bank of stimuli (separate bank for each graph). Each graph was learnt during four blocks (Fig. 1b; 4 blocks for graph 1 followed by 4 blocks for graph 2). Participants could take short resting breaks during the blocks. They were instructed to take a longer resting break after completing learning the first graph. A third graph was learnt on the second day during seven blocks of the task. Data analysis is for all second-day trials.

**Block structure**. The structure of each experiment block in each experiment and day is outlined in Tables 1–2 below (the order of tasks in a block corresponds to

**Table 1 Transferring of hexagonal structure.**

| Task name | Day 1 | Day 2 |
|---|---|---|
| Learning phase | Random walk | Pairs |
| Extending pictures sequences | Yes | Yes |
| Can it be in the middle | Yes | Yes |
| Navigation | Yes | No |
| Distance estimation | Yes | Yes |

**Table 2 Transferring of community structure.**

| Task name | Day 1 | Day 2 |
|---|---|---|
| Learning phase | Random walk | Random walk |
| Extending pictures sequences | Yes | Yes |
| Can it be in the middle | Yes | Yes |
| Navigation | Yes | Yes |
| Distance estimation | Yes | Yes |

Note: "Yes" and "No" refer to the inclusion of a task in an experimental block.

moving from the top to the bottom of the corresponding table). Next, we elaborate the various components of each block.

*Learning phase:* We used different protocols for the learning phases of experimental blocks as follows:

(1) In the "Random walk" protocol participants learned associations between graph nodes by observing a sequence of pairs of pictures which were sampled from a random walk on the graph (successive pairs of pictures shared a common picture). Participants were instructed to 'say something in their head' in order to remember the associations. Hexagonal graphs included 120 steps of the random walk per block and community-structured graphs included 180 steps per block (we introduced more pictures in the community graph condition as random walks on such graphs result in high sampling of transitions within a certain community and low sampling of transitions between communities).

(2) In the "Pairs" protocol participants learned the associations between graph nodes by observing pairs of pictures. Each pair of pictures corresponds to two neighbouring nodes (i.e., an edge) on the graph. Some edges were excluded from the graph ("missing links"), otherwise, the pairs were sampled uniformly randomly according to a uniform distribution and independently across pairs. 150 pairs were presented in each block (with repetition).

The reason we used the "pairs" protocol for Day 2 of Exp. 1 is as follows: Exp. 1 was designed to test participants' ability to infer missing graph links (edges). However, a link that is constantly missing may lead to an inference of the existence of an obstacle rather than an unobserved link. We speculated that learning by sampling pairs of neighbouring nodes, instead of learning from pairs that are taken from random walks on the graph, would reduce this risk. Following the same reasoning, we excluded the navigation task (described below) during the second day of that experiment (hexagonal condition only), as navigation necessarily involves walks on the graph (see "Navigation" rows of Tables 1 and 2).

*Extending pictures sequences:* Given a target picture, which of two sequences of three pictures can be extended by that picture (a sequence can be extended by a picture only if it is a neighbour of the last picture in the sequence, the correct answer can be sequence 1/sequence 2/both sequences): Sixteen questions per block. (A picture could not appear twice in the same sequence, therefore, if the target picture is already in the sequence the correct answer was necessarily the other sequence).

*Can it be in the middle:* Determine whether a picture can appear between two other pictures, the answer is yes if and only if the picture is a neighbour of the two other pictures. Sixteen questions per block.

*Navigation task*: The aim—navigating to a target picture. Participants are informed that they are currently at the picture that appears on the left of the screen. They were asked to choose between two pictures that are associated with that picture or skip and sample again two pictures that are associated with the current picture (skip is counted as a step). On each step participants were instructed to choose a picture that they think has a smaller number of links to the target picture (according to their memory). Following their choice, the chosen picture appeared on the left and two new pictures, that correspond to states that are neighbours of the chosen picture, appear in the middle (Fig. 1b). Once a participant selected a neighbour of the target picture, the target picture itself can appear as a picture that can be chosen. The game terminated when either the target was reached or 200 steps were taken (without reaching the target). In the latter case a message 'too many steps' was displayed. On the first block, the number of links from the current picture to the target picture was shown on the screen. Participants played three

games in each block. The starting distance (number of links) between the starting picture to the target was 2, 3 and 4.

*Distance estimation:* Which of two pictures has the smallest number of links to a target picture: 45 questions per block.

**Statistical values**. *First experiment* (Fig. 5)

All questions: $p$-value = 0.0068, $t$ = 2.54, sd = 0.0547, ci = [0.0124 inf], $d$ = 0.699

Inference questions: $p$-value = 0.016, $t$ = 2.19, sd = 0.066, ci = [0.0088 inf], $d$ = 0.565

One-tailed $t$ test, df = 58. The results are significant for two-tailed test as well. *Second experiment* (Fig. 6)

*Response time (Learning pace):* $p$-value = 0.003, $t$ = 3.19, sd = 0.5, ci = [0.18, 0.83], $d$ = 1.01 (two-tailed, df = 38).

Correct structural prior leads to faster navigation to the target:

Number of steps to the target is two, $p$-value = 0.005, $t$ = −2.68, sd = 14.58, ci = [−inf, −4.6], $d$ = 0.85.

Number of steps to the target is three, $p$-value = 0.026, $t$ = −2.02, sd = 14.85, ci = [−inf, −1.48], $d$ = 0.63.

Number of steps to the target is four, $p$-value = 0.006, $t$ = −2.6, sd = 10.54, ci = [−inf, −3.04], $d$ = 0.82.

One-tailed $t$ test, df = 38.

Choose connecting nodes:

All answers: $p$-value = 0.03, $t$ = 2.25, sd = 0.1, ci = [0.007,0.13], d = 0.71 (two-tailed, df = 38)

Incorrect answer: $p$-value = 0.006, $t$ = 2.88, sd = 0.1, ci = [0.03,0.16], d = 0.91 (two-tailed, df = 38)

ci is the confidence interval and $d$ is Cohen $d'$ (effect size).

**Reporting summary**. Further information on research design is available in the Nature Research Reporting Summary linked to this article.

## Data availability

The authors declare that the data supporting the findings of this study are available within the paper and its Supplementary information files. Source data are provided with this paper.

## Code availability

The authors declare that the codes for the simulations in this study are available within the paper and its Supplementary information files.

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

## Acknowledgements

We thank Alon Baram and David McCaffary for careful reading of the manuscript and Tamas Madarasz for his support in addressing theoretical questions. This research was funded by: Wellcome Trust Senior Research Fellowship (104765/Z/14/Z) and Prinicpal Research Fellowship (219525/Z/19/Z), together with a James S. McDonnell Foundation Award (JSMF220020372), to T.E.J.B.

## Author contributions

S.M., S.W.K. and T.E.J.B. conceived the study. S.M. and T.E.J.B. designed the experiment. S.M. programmed the experiments. S.M. performed the experiments. S.M. and T.E.J.B. developed the models. R.M. and T.P. advise on the model. S.M. implemented the model. S.M., S.W.K. and T.E.J.B. wrote the manuscript. All authors discussed the results and commented on the paper.

## Competing interests

The authors declare no competing interests.
