## [Peer Review File · Nature Communications]

Reviewers' Comments:

Reviewer #1:

Remarks to the Author:

Review of "Transferring structural knowledge across cognitive maps in humans and models"

This manuscript describes the results of a set of two behavioral studies in which human participants performed on a structure learning task whereby sequences of stimuli are presented such that the relation between stimuli are governed by distinct graph structures: either a hexagonal graph or one with community structure. On day 1 participants learn one graph structure, and on the 2nd day they are tested with another that is either concordant or distinct from the one they learned on the 1st day, therefore enabling the authors to test for transfer of structure knowledge between graphs. The authors also present a hidden markov model, in which the graph structure is encoded as an abstract basis set informed by the size of the graph. Which graph applies in a given context is then inferred, alongside graph size and the mapping between stimuli and graph nodes. The authors present evidence that participants perform better on the 2nd day on various knowledge tests of graph structure if they learned on the same structure (e.g. hex) compared to the wrong structure on the 1st day. This includes knowledge transfer for parts of the graph that they did not experience directly on the 1st day but could merely infer. The authors conclude that humans are capable of generalizing structure knowledge, including to unobserved state-trajectories.

Overall, this manuscript describes a promising and interesting line of work about the extent to which humans are capable of learning structure in their environment, internalizing that structure in terms of a cognitive map, and applying that structure to new situations. In that sense, the manuscript provides insight into a novel and poorly understood aspect of human reinforcement-learning.

I have some comments:

(1) Overall the manuscript is very hard to read. For instance, different terms are used to describe the same phenomena in ways that are sometimes inconsistent. For example, the terms conjunctive vs abstract representation is used in Figure 2, but those terms are not described or discussed in the main text. I know what they are trying to show here, but it took me a while to try to understand what the figure was about. In figure 1, the way the two experiments are concatenated and the different tests are described is also confusing – so far as I can tell some tests were run on experiment 2 (e.g. navigation) and others only on experiment 1 e.g. distance estimation? But maybe I'm wrong here. I think clearly separating the descriptions of experiment 1 and 2 in the text and having separate figures for exactly what was done in both might help a lot. Trying to understand exactly what was done in experiment 1 and 2 took me as a reader a lot longer than it should have. The methods section also seems hastily written. Some parts of the methods are written in abbreviated form which gives the impression the authors didn't have time to finish the manuscript before submitting it. There are also typos (e.g. participnats, smapling). Simple details are also missing -- .e.g. even minimal demographic details (mean age, gender) of participants are missing. The authors need to go through the manuscript and rewrite it to improve clarity and detail in presentation throughout.

(2) The authors propose a HMM model as a way of solving this task. They hint that other model frameworks would fail on this task. For instance, in figure 2, they allude to an associative learning model that would fail to enable inference (presumably) and/or generalization. They also allude to the fact that a successor state representation learning model would also fail on this. I agree that this is the case. But, it would be nice to see clear examples of where and how such models would fail, and how the model the authors propose can clearly outperform these other models. This may be

somewhat trivial, but it would be nice to outline why such models would fail more explicitly. Perhaps this would be mostly for didactic purposes – but I think it's important to do so nonetheless, at minimum to improve clarity of presentation of the model, so the reader can understand what is different in what is being proposed here compared to the standard in the field.

Furthermore, it is usually good practice to present competing models that even have a chance at succeeding on a task like this, in which there is some attempt at a model comparison against the human data.

Model comparison notwithstanding, I think it is imperative that the authors try to link their specific proposed model to the actual human behavior more seriously. As it is, the authors present behavioral data supporting the idea that participants encode structure knowledge and are capable of using it in various ways. However, are there specific predictions that come from this model about HOW participants would be expected to perform on this task on a trial by trial or block by block basis? For instance over the course of learning, would the authors expect particular patterns in the behavior to evolve over time as participants are converging on the correct structure? If participants are given the wrong structural priors, are particular error patterns expected compared to if they had no structural priors whatsoever? What I'm trying to get at here is the need for a more serious attempt to examine the model vs human behavior and find situations where the model makes particular predictions that are confirmed in the data (or not). No model is perfect, so it would be equally interesting to see predictions that are not confirmed, as well as ones that are confirmed.

(3) Relationship between performance on day 1 and day 2. The authors note that for the inference over unobserved links, only some participants could perform such an inference. This is not surprising as individual variation would be expected. However, it would be nice to see more careful consideration of the relationship between learning and performance on day 1 and day 2. Are the people who did the inference task well on day 2, also ones who performed well on day 1? In other words, if I haven't learned well on day 1, I have no chance of showing structure knowledge on day 2. Getting some handle on what is the difference between people who do well on the task on day 2 compared to day 1 would be super useful in that it could further support the authors' claims – both for the inference specifically as well as for task performance more generally. Also, is there any way we can understand what the people who do poorly on the structure inference questions are doing? Is their behavior consistent with another cognitive strategy or are they just behaving randomly?

Reviewer #2:

Remarks to the Author:

This paper presents a theoretical and experimental analysis of structural transfer in a graph learning task. The authors propose a model of structure learning that predicts transfer of structural knowledge, which they test in human participants.

Overall, I thought this was an interesting paper, which contributes usefully to the literature on structure learning and transfer. The clarity could be improved and a more thorough empirical argument for the model needs to be made.

Major comments:

The description of the experiments needs more detail. I couldn't find any description of what the graphs were or how they were constructed. Some graphs are shown in Figure 1, but as I understand it the participants learned multiple graphs of each type, which aren't shown.

The two experiments are in some sense symmetrical, really two conditions of a single experiment. I don't care about the fact that they are described as two different experiments, but what seems more important is the lack of symmetry in the analyses of the data. Why weren't the same analyses applied to both experiments? Have I misunderstood something about the setup that would preclude this?

There's no direct comparison between the model and experimental data. Ideally we would see the same analyses applied to both model and human participants, plotted in the same way. There is already an indication in the figures that the model is much better than people at this task. The authors should address this mismatch.

There are no comparisons with alternative models that could conceivably solve the same task. For example, Kemp & Tenenbaum (2008) propose a different (and much richer) generative model for graphs. Or within the class of models considered in this paper, what about alternative basis sets? Does the particular choice of basis set made here predict a distinctive pattern of generalization (compared to other choices)? It's also worth pointing out that the authors restricted the structural forms to only the classes that appeared in their experiment, whereas more generally the space is presumably much broader (as argued by Kemp & Tenenbaum).

The authors have done a good job connecting their work to ideas in machine learning and neuroscience, but apart from Tolman there is very little discussion of relevant work in psychology. Below I've given a few references that explore relational/causal transfer. I'm not sure all of these citations are needed; I will leave this to the authors' discretion depending on how they want to discuss this research.

- Halford et al. (1998), *Cognitive Psychology*
- Honey & Watt (1998), *JEP:ABP* [this work stimulated a line of research on acquired relational equivalence]
- Kemp et al. (2010), *Cognitive Science*
- Lu et al. (2016), *Cognitive Science*
- A large literature on analogical reasoning. For a review of the early work in this area, see Reeves & Weisberg (1994), *Psychological Bulletin*.

Minor comments:

p. 3: "a learning phase, following" -> "a learning phase. Following"

Fig 3 caption: "that determined" -> "that is determined"

p. 7: what statistical test is being reported for the data in Fig 4b?

p. 8: why is the t-test one-tailed? Same question for all the other one-tailed tests reported in the paper. Sometimes the tests are two-tailed, with no explanation of why one-tailed vs. two-tailed is chosen. I think the default should be two-tailed.

p. 8: how was the high-performing subset of participants identified?

p. 8: "These effects cannot be driven by non-inferential approximations of graph distance". I don't think the relevant issue is about inference vs. non-inference, but rather about the generalization abilities of different models. If you parametrized a successor representation with a basis set that can generalize across graphs, then I see no reason why it couldn't in principle do this task.

p. 16: "inferece" -> "inference"

p. 17: "a basis sets" -> "a basis set"

p. 17: "using standard interpolating method" -> "using a standard interpolating method" [I actually don't know what standard interpolation method applies to eigenvector resizing]

p. 17: "Participnats" -> "Participants"

p. 17: "taking from" -> "taken from"

p. 17: why is "bank" capitalized?

p. 18: "Madaresz" -> "Madarasz"

Reviewers' comments- blue

Our response – black

Text from revised manuscript – green

Reviewers comments:

Reviewer #1 (Remarks to the Author):

Review of “Transferring structural knowledge across cognitive maps in humans and models”

This manuscript describes the results of a set of two behavioral studies in which human participants performed on a structure learning task whereby sequences of stimuli are presented such that the relation between stimuli are governed by distinct graph structures: either a hexagonal graph or one with community structure. On day 1 participants learn one graph structure, and on the 2nd day they are tested with another that is either concordant or distinct from the one they learned on the 1st day, therefore enabling the authors to test for transfer of structure knowledge between graphs. The authors also present a hidden markov model, in which the graph structure is encoded as an abstract basis set informed by the size of the graph. Which graph applies in a given context is then inferred, alongside graph size and the mapping between stimuli and graph nodes. The authors present evidence that participants perform better on the 2nd day on various knowledge tests of graph structure if

they learned on the same structure (e.g. hex) compared to the wrong structure on the 1st day. This includes knowledge transfer for parts of the graph that they did not experience directly on the 1st day but could merely infer. The authors conclude that humans are capable of generalizing structure knowledge, including to unobserved state-trajectories.

Overall, this manuscript describes a promising and interesting line of work about the extent to which humans are capable of learning structure in their environment, internalizing that structure in terms of a cognitive map, and applying that structure to new situations. In that sense, the manuscript provides insight into a novel and poorly understood aspect of human reinforcement-learning.

I have some comments:

(1) Overall the manuscript is very hard to read.

Thanks we agree. We have made substantive changes to the structure of the manuscript. We hope these changes clarify the logic of the modelling and therefore address some of the questions below, but we also think they make the manuscript much clearer and easier to read. As well as this, we have addressed the particular suggestions below.

The structure of the manuscript now runs as follows:

We present two different ways to represent a cognitive map. The first relies on associative learning. This type of learning results in a conjunctive representation of the structure of the graph and the stimuli representation. We considered different models of such learning: learning the transition matrix, learning successor representation (SR) and a filtered version of the SR. We then consider a different type of representation, in which the structure of the task is represented abstractly and the associations between the abstract states and the stimuli are learnt.

We show that simple associative models cannot account for inference of unobserved links while a filtered version of such representation can. We further show that transfer of abstract representation of the structure of the task also enables inference of unobserved links.

This sets up the experimental portion of the manuscript. The two key questions are (1) can subjects infer links and (2) does this capacity transfer across graphs. Here we show that the prior structural knowledge participants have is crucial for their success in predicting the existence of unobserved links. This experiment leads us to the conclusion that abstract representation of the structure of the task exists in human cognition. This abstract representation allows participants to infer the unobserved links.

We then present a second experiment that shows how abstract representation of the structure can lead to different learning and behavioural policies that are relevant to the abstract structural form.

We hope that this clarifies the purpose of the modelling (more on this below). We are not trying to fit individual data. We are using modelling to make explicit the logic of the experiment.

A large amount of text has changed in the manuscript to reflect this structural change. We will copy here the main additions:

We added the following paragraphs in results:

Associative representation

Learning such graphs can be accomplished using different types of representations. One solution to such a problem is a conjunctive representation of the stimuli and their relationships; the relationships between the stimuli are encoded by the associations between the representations of the stimuli themselves (figure 2). An example of such representation is the Successor Representation (SR)²⁵. Here the representation of each state (in our setup each stimulus defines a state) encodes the probability to reach any other states in the future. Using such a representation it is possible to determine whether two stimuli are neighbouring nodes on a graph and even to navigate on a graph.

*In : “**Inferring unobserved trajectories**” we have added the following paragraph (beginning of paragraph 3) :*

Simple associative models, such as learning the transition matrix between the pictures themselves or learning SR, cannot solve such a problem (Figure 4a). Following Stachenfeld et

a^2 , we spectrally filtered the SR that is currently being learnt using its own eigen-decomposition; we reconstructed the SR using the seven most informative eigenvectors only. Such filtration smooth over the unobserved links, which then allows the agent to answer such questions better than chance without the need of knowledge transfer (Figure 4a, $p < 0.001$ one tailed ttest).

In the same section, we have added another paragraph at the end:

We then wanted to check whether humans solve such problems using transfer of abstract structural knowledge or whether they exploit a smoothed associative representation. If participants can solve the task without the need of prior knowledge it will imply that associative learning is enough, while if their performance depends on prior knowledge, we can conclude that humans do represent and transfer abstract structural knowledge. To test whether humans infer unobserved edges by using clever smoothing of noisy representation or whether an abstract knowledge is being transferred, we designed the following two experiments.

We have changed and spilt figure 4 into:

Figure 4: inference of unobserved links (Hexagonal graph)

A) Inferring the existence of unobserved edges (links). Left – the task: The agents had to indicate which of two nodes (pictures) has smaller number of links to the target. With only observed links, the number of links to the target was identical. Right – red edges indicates missing links on the graph. For example, the two nodes that are marked with light blue have the same number of observed links to the target node (marked with

dark blue circle), while the number of links that connect these two nodes to the target is different on the complete graph.

- B) When learning from pairs that were sampled randomly (not in succession) while some of the links (pairs) were never observed, simple associative models as learning transition matrix (DA) or simple SR (SR-online: learning using TD-SR²⁸, SR-A: calculating SR from the learnt transition matrix) could not infer the existence of the unobserved links and solve the task (it in fact solves it worse than chance). Agents that use a filtered SR representation (SRreg) could answer these questions better than chance. Shadows are the standard errors.
- C) When learning from pairs that were sampled randomly (not in succession) while some of the links (pairs) were never observed), the basis set agent, that transfers abstract structural knowledge, was able to infer the structural form (Figure S2) and graph size correctly (upper panel). Further, the agent was able to infer the existence of links that were never observed and determined correctly, which of two pictures is closer to a target picture, according to the complete graph (green). The agent could do so even though the number of observed links between the two pictures and the target was identical ($p(\text{cor})$ corresponds to the average fraction of correct answers out of 40 questions in each block). When the agent was forced to infer a community structure (red), it answered these questions worse than chance. Shadows are the standard errors.

Figure 5: Transfer of structural knowledge allows inference of unobserved links (Hexagonal graph)

Participants had to indicate which of two pictures is closer to a target picture. Participants that reach the second day of our task with the correct prior expectation over the structural form performed significantly better in such task compared to participants with the wrong structural prior (left panel). (30 participants in each group). They were able to answer these questions significantly above chance even when there were links that were never observed, and they had to choose between two

pictures with identical number of observed links to the target (right panel). Error bar: SEM. Colorcode: $\text{Log}_{10}(p_{\text{value}})$

For instance, different terms are used to describe the same phenomena in ways that are sometimes inconsistent. For example, the terms conjunctive vs abstract representation is used in Figure 2, but those terms are not described or discussed in the main text. I know what they are trying to show here, but it took me a while to try to understand what the figure was about.

We replaced the term ‘conjunctive representation’ with ‘associative representation’ and have added the paragraph as written above.

In figure 1, the way the two experiments are concatenated and the different tests are described is also confusing – so far as I can tell some tests were run on experiment 2 (e.g. navigation) and others only on experiment 1 e.g. distance estimation? But maybe I’m wrong here. I think clearly separating the descriptions of experiment 1 and 2 in the text and having separate figures for exactly what was done in both might help a lot. Trying to understand exactly what was done in experiment 1 and 2 took me as a reader a lot longer than it should have.

Thanks, we did have them separated in the original paper, but we can see where the confusion came from. We have now made it completely explicit. We have added a table in the methods section that describes which tasks were done in each experiment and during each day. We have added the following sentence to the caption of figure 1: ‘Question type three was excluded from day 2 on experiment 1’.

The tables:

Table 1: Transferring of Hexagonal structure

Task Name	Day 1	Day2
Learning phase	Random walk	Pairs
Extending pictures sequences	yes	yes
Can it be in the middle	yes	yes
Navigation	yes	No
Distance estimation	yes	yes

Table 2: Transferring of Community structure

Task Name	Day 1	Day2
Learning phase	Random walk	Random
Extending pictures sequences	yes	yes
Can it be in the middle	yes	yes
Navigation	yes	Yes
Distance estimation	yes	yes

The methods section also seems hastily written. Some parts of the methods are written in abbreviated form which gives the impression the authors didn't have time to finish the manuscript before submitting it. There are also typos (e.g. participnats, smapling). Simple details are also missing -- .e.g. even minimal demographic details (mean age, gender) of participants are missing. The authors need to go through the manuscript and rewrite it to improve clarity and detail in presentation throughout.

We have rewrite the method section. Main additions:

Successor Representation Model

The successor representation is defined as:

$$SR = \sum_t \gamma^t A^t = inv(I - \gamma A)$$

Where A is the transition matrix, I is the identity matrix and γ is a discount factor.

The SR can be updated within blocks, after updating the transition matrix online, or using TD learning²⁸. After observing a transition at time step $t+1$ of $s_t \rightarrow s_{t+1}$ the SR is updated according to:

$$SR_{t+1}(s_{t+1}, s') = SR(s_t, s') + \alpha [I(s_t = s') + \gamma SR(s_{t+1}, s') - SR(s_t, s')]$$

Where α is the learning rate. We made the SR symmetrical at the end of each block.

Spectral regularization (filtering the SR using its own eigendecomposition):

We have calculated the eigendecomposition (using SVD) of the SR that has been learnt using TD. We then calculated the regularized SR by: $SR_s = U_m S_m U_m^T$, where U_m is the matrix of the $m=7$ most informative eigenvectors of the symmetrised SR and S_m is a diagonal matrix with the $m=7$ largest eigenvalues on its diagonal. The number of simulations for figure 4a is 10.

Behavioural experiments

Participants: We recruited 100 participants, 60 participants for experiment 1 (30 in each group) and 40 participants for experiment 2 (20 in each group). All participants are UCL students with an average age of 23.5.

The study was approved by the University College London Research Ethics Committee (Project ID 11235/001). Participants gave written informed consent before the experiment.

Graphs structure:

Experiment 1: transfer of hexagonal structure: Each hexagonal graph consisted of 36 nodes and periodic boundary conditions as shown in Fig.1

Experiment 2: transfer of community structure: Each graph consisted of five communities with seven nodes each. Within a community, each node was connected to all other nodes except for connecting nodes that were not connected to each other but were each connected to a connecting node of a neighbouring community (Fig. 1). Therefore all nodes had a degree of six, similarly to hexagonal graphs. Our community-structure graph had an hierarchical structure, wherein communities are organised on a ring. We hypothesized that inference of the second order structure of a ring and transfer of this structure from day one to day two will allow participants to infer a missing link that closes the ring. We therefore introduced a missing link during the second day (see supplementary for the results).

Experimental Procedures

Participants learned two graphs with the same underlying structure but different stimuli during the first day. Stimuli were selected randomly, for each participant, from a bank of stimuli (separate bank for each graph). Each graph was learnt during four blocks (figure 1b; 4 blocks for graph 1 followed by 4 blocks for graph 2). Participants could take short resting breaks during the blocks. They were instructed to take a longer resting break after completing learning the first graph. A third graph was learnt on the second day during seven blocks of the task. Data analysis is for all second day trials.

Block structure:

The structure of each experiment block in each experiment and day is outlined in Tables 1-2 below (the order of tasks in a block corresponds to moving from the top to the bottom of the corresponding table). Next, we elaborate the various components of each block.

Table 1: Transferring of hexagonal structure

Task Name	Day 1	Day2
Learning phase	Random walk	Pairs
Extending pictures sequences	yes	yes
Can it be in the middle	yes	yes
Navigation	yes	No

Distance estimation	yes	yes
-----	-----

Table 2: Transferring of Community structure

Task Name	Day 1	Day2
Learning phase	Random walk	Random Walk
Extending pictures sequences	yes	yes
Can it be in the middle	yes	yes
Navigation	yes	Yes
Distance estimation	yes	yes

Note: “Yes” and “No” refer to the inclusion of a task in an experimental block.

Learning Phase: We used different protocols for the learning phases of experimental blocks as follows:

- 1) In the “Random walk” protocol participants learned associations between graph nodes by observing a sequence of pairs of pictures which were sampled from a random walk on the graph (successive pairs of pictures shared a common picture). Participants were instructed to ‘say something in their head’ in order to remember the associations. Hexagonal graphs included 120 steps of the random walk per block and community structured graphs included 180 steps per block (we introduced more pictures in the community graph condition as random walks on such graphs result in high sampling of transitions within a certain community and low sampling of transitions between communities).
- 2) in the “Pairs” protocol participants learned the associations between graph nodes by observing pairs of pictures. Each pair of pictures corresponds to two neighbouring nodes (i.e., an edge) on the graph. Some edges were excluded from

the graph (“missing links”), otherwise, the pairs were sampled uniformly randomly according to a uniform distribution and independently across pairs. 150 pairs were presented in each block (with repetition).

The reason we used the “pairs” protocol for Day 2 of Exp. 1 is as follows: Exp.1 was designed to test participants’ ability to infer missing graph links (edges). However, a link that is constantly missing may lead to an inference of the existence of an obstacle rather than an unobserved link. We speculated that learning by sampling pairs of neighbouring nodes, instead of learning from pairs that are taken from random walks on the graph, would reduce this risk. Following the same reasoning, we excluded the navigation task (described below) during the second day of that experiment (hexagonal condition only), as navigation necessarily involves walks on the graph (See “Navigation” rows of Tables 1 and 2”).

Extending pictures sequences: Given a target picture, which of two sequences of three pictures can be extended by that picture (a sequence can be extended by a picture only if it is a neighbour of the last picture in the sequence, the correct answer can be sequence 1/sequence 2/ both sequences): 16 questions per block. (A picture could not appear twice in the same sequence, therefore, if the target picture is already in the sequence the correct answer was necessarily the other sequence).

Can it be in the middle: Determine whether a picture can appear between two other pictures, the answer is yes if and only if the picture is a neighbour of the two other pictures. 16 questions per block.

navigation task: The aim – navigating to a target picture. Participants are informed that they are currently at the picture that appears on the left of the screen. They were asked to choose between two pictures that are associated with that picture or skip and sample again two pictures that are associated with the current picture (skip is counted as a step). On each step participants were instructed to choose a picture that they think has a smaller number of links to the target picture (according to their memory). Following their choice, the chosen picture appeared on the left and two new pictures, that correspond to states that are neighbours of the chosen picture, appear in the middle (Figure 1b). Once a participant selected a neighbour of the target picture, the target picture itself can appear as a picture that can be chosen. The game terminated when either the target was reached or 200 steps were taken (without reaching the target). In the latter case a message ‘too many steps’ was displayed. On the first block, the number of links from the current picture to the target picture was shown on the screen. Participants played three games in each block. The starting distance (number of links) between the starting picture to the target was 2, 3 and 4.

Distance estimation: Which of two pictures has the smallest number of links to a target picture: 45 questions per block.

Statistical values

All questions: p-value = 0.0068, t = 2.54, sd = 0.0547, ci = [0.0124 inf], d = 0.699

Inference questions: p-value = 0.016, t = 2.19, sd = 0.066, ci = [0.0088 inf], d = 0.565

One-tailed ttest, df = 58.

Second experiment (Figure 6)

Response Time (Learning pace): p -value = 0.003, t = 3.19, sd = 0.5, ci = [0.18, 0.83], d = 1.01 (two-tailed, df = 38).

Correct structural prior leads to faster navigation to the target:

Number of steps to the target is two, p -value = 0.005, t = -2.68, sd = 14.58, ci = [-inf,-4.6], d = 0.85.

Number of steps to the target is three, p -value = 0.026, t = -2.02, sd = 14.85, ci = [-inf,-1.48], d = 0.63.

Number of steps to the target is four, p -value = 0.006, t = -2.6, sd = 10.54, ci = [-inf, -3.04], d = 0.82.

One-tailed t test, df = 38.

Choose connecting nodes:

All answers: p -value = 0.03, t = 2.25, sd = 0.1, ci = [0.007,0.13], d = 0.71 (two-tailed, df = 38.)

Incorrect answer: p -value = 0.006, t = 2.88, sd = 0.1, ci = [0.03,0.16], d = 0.91 (two-tailed, df = 38.)

ci is the confidence interval and d is Cohen d' (effect size).

(2) The authors propose a HMM model as a way of solving this task. They hint that other model frameworks would fail on this task. For instance, in figure 2, they allude to an associative learning model that would fail to enable inference (presumably) and/or generalization. I agree that this is the case. But, it would be nice to see clear examples of where and how such models would fail, and how the model the authors propose can clearly outperform these other models. This may be somewhat trivial, but it would be nice to outline why such models would fail more explicitly. Perhaps this would be mostly for didactic purposes – but I think it's important to do so nonetheless, at minimum to improve clarity of presentation of the model, so the reader can understand what is different in what is being proposed here compared to the standard in the field. Furthermore, it is usually good practice to present competing models that even have a chance at succeeding on a task like this, in which there is some attempt at a model comparison against the human data.

We are very thankful for this suggestion. We have changed the manuscript structure, as described above by adding different associative models and checking their success in inferring unobserved links. We copied the relevant additions in our response to comment (1).

Model comparison notwithstanding, I think it is imperative that the authors try to link their specific proposed model to the actual human behavior more seriously. As it is, the authors present behavioral data supporting the idea that participants encode structure knowledge and are capable of using it in various ways.

However, are there specific predictions that come from this model about HOW participants would be expected to perform on this task on a trial by trial or block by block basis? For

instance over the course of learning, would the authors expect particular patterns in the behavior to evolve over time as participants are converging on the correct structure? If participants are given the wrong structural priors, are particular error patterns expected compared to if they had no structural priors whatsoever?

What I'm trying to get at here is the need for a more serious attempt to examine the model vs human behavior and find situations where the model makes particular predictions that are confirmed in the data (or not). No model is perfect, so it would be equally interesting to see predictions that are not confirmed, as well as ones that are confirmed.

We disagree with this perspective, and hope that we can persuade the reviewer of our perspective. We are not trying to make a model that predicts human behaviour quantitatively. In order to make such a model, we would have to account for the failures of human memory; For the attentional lapses; for the speed of learning. Etc. etc. It would be a highly parameterised model. In our view, such a model would not likely add any conceptual insight over and above the behavioural effects themselves.

Instead, the purpose of our model is (as the reviewer states for the associative models above), illustrative and didactic. It is a model that teaches us what the critical experiment is to perform. It shows that we should test for inference, and transfer. The modelling teaches us that any evidence of inference rules out simple SR associative models, and any evidence of transfer implies abstraction.

This is the role of the modelling in this manuscript. We understand that this is different from the role of modelling in other manuscripts that are, for example, trying to extract learning rates from human participants. We believe modelling can play both roles, and we hope the reviewer can agree that the role it is playing in this manuscript is valuable.

(3) Relationship between performance on day 1 and day 2. The authors note that for the inference over unobserved links, only some participants could perform such an inference. This is not surprising as individual variation would be expected. However, it would be nice to see more careful consideration of the relationship between learning and performance on day 1 and day 2. Are the people who did the inference task well on day 2, also ones who performed well on day 1? In other words, if I haven't learned well on day 1, I have no chance of showing structure knowledge on day 2. Getting some handle on what is the difference between people who do well on the task on day 2 compared to day 1 would be super useful in that it could further support the authors' claims – both for the inference specifically as well as for task performance more generally.

We thank the reviewer for this suggestion. We agree that this would have been an interesting finding. However, there is no evidence for such a correlation in the data. We have added the analysis to the supplementary information.

Also, is there any way we can understand what the people who do poorly on the structure inference questions are doing? Is their behavior consistent with another cognitive strategy or are they just behaving randomly?

To our understanding, they are performing randomly as their results are approximately 50%. We don't have good suggestions for how to analyse this data to look for alternative cognitive strategies.

Reviewer #2 (Remarks to the Author):

This paper presents a theoretical and experimental analysis of structural transfer in a graph learning task. The authors propose a model of structure learning that predicts transfer of structural knowledge, which they test in human participants.

Overall, I thought this was an interesting paper, which contributes usefully to the literature on structure learning and transfer. The clarity could be improved and a more thorough empirical argument for the model needs to be made.

*Thanks very much for these comments. We have restructured the manuscript substantially, which we hope will make it clearer. Both yourself and R1 raised issues with the modelling section. In essence, the role of the model in the manuscript was not clear. We have tried to clarify that by adding alternative models that do not abstract the graph structure. This allows us to demonstrate in the modelling section what **qualitative effects** we need to show in the data. In essence, we show that to demonstrate that graph structure is abstracted, it is not sufficient merely to show inference, we need to show **inference and transfer**. That is, we need to show a **difference in link inference between groups with different priors**. This is the role of the modelling in the paper (together with a suggestion of a brain-inspired mechanism for representing abstract structural knowledge) and we have tried to clarify this in the new manuscript.*

*Below, you have made suggestions that we take a more **quantitative approach** to the behavioural modelling. For reasons that we explain below, we have opted not to do this. We hope that you will agree with us that this is reasonable given the role of the modelling in the paper described in the previous paragraph. The model is not intended to be a detailed model of subject behaviour (nor is it close to being one for reasons we give in response to your questions below). Instead, the model serves to demonstrate the key **qualitative tests** that are required to demonstrate abstraction. We then look for these qualitative effects in the data.*

Major comments:

The description of the experiments needs more detail. I couldn't find any description of what the graphs were or how they were constructed. Some graphs are shown in Figure 1, but as I understand it the participants learned multiple graphs of each type, which aren't shown.

We thank the reviewer for this comment. Figure 1 has been changed as well as the text in the beginning of the results. We rewrite the methods section and include a table that describes the tasks participants have done during each day on each experiment.

Figure 1: Transfer of structural knowledge: Graph structures and experimental design.

A) Experimental design. Agents and participants learned graphs with underlying Hexagonal (left) or Community (right) structure. Each grey dot is a node on a graph and corresponds to a picture that was viewed by the participant (for example, a picture of an ice-cream). The lines are edges between nodes. Pictures of nodes that are connected by an edge can appear one after the other. The degree of all nodes in both graphs is six (a connecting node connects to one fewer nodes within a community to keep the degree six). Participants learned the graphs during two successive days. In both experiments, participants were segregated into two groups. Participants of one group learned graphs with the same underlying structure during both days while the other groups learned graphs with different underlying structures during the different days. Two graphs were learnt during day 1 and additional graph on day 2.

we have added the following sentences at the end of the first paragraph in the result section: During the first day, participants learned two different graphs, with different pictures set but same structure, while during the second day participants learned a third graph with new pictures set (one group learned a graph with the same underlying structure and the other group learned a graph with a different structure, figure 1).

new experimental details in the method section:

Behavioural experiments

Participants: We recruited 100 participants, 60 participants for experiment 1 (30 in each group) and 40 participants for experiment 2 (20 in each group). All participants are UCL students with an average age of 23.5.

The study was approved by the University College London Research Ethics Committee (Project ID 11235/001). Participants gave written informed consent before the experiment.

Graphs structure:

Experiment 1: transfer of hexagonal structure: Each hexagonal graph consisted of 36 nodes and periodic boundary conditions as shown in Fig.1

Experiment 2: transfer of community structure: Each graph consisted of five communities with seven nodes each. Within a community, each node was connected to all other nodes except for connecting nodes that were not connected to each other but were each connected to a connecting node of a neighbouring community (Fig. 1). Therefore all nodes had a degree of six, similarly to hexagonal graphs. Our community-structure graph had an hierarchical structure, wherein communities are organised on a ring. We hypothesized that inference of the second order structure of a ring and transfer of this structure from day one to day two will allow participants to infer a missing link that closes the ring. We therefore introduced a missing link during the second day (see supplementary for the results).

Experimental Procedures

Participants learned two graphs with the same underlying structure but different stimuli during the first day. Stimuli were selected randomly, for each participant, from a bank of stimuli (separate bank for each graph). Each graph was learnt during four blocks (figure 1b; 4 blocks for graph 1 followed by 4 blocks for graph 2). Participants could take short resting breaks during the blocks. They were instructed to take a longer resting break after completing learning the first graph. A third graph was learnt on the second day during seven blocks of the task. Data analysis is for all second day trials.

Block structure:

The structure of each experiment block in each experiment and day is outlined in Tables 1-2 below (the order of tasks in a block corresponds to moving from the top to the bottom of the corresponding table). Next, we elaborate the various components of each block.

Table 1: Transferring of hexagonal structure

Task Name	Day 1	Day2
Learning phase	Random walk	Pairs
Extending pictures sequences	yes	yes
Can it be in the middle	yes	yes
Navigation	yes	No

Distance estimation	yes	yes
-----	-----

Table 2: Transferring of Community structure

Task Name	Day 1	Day2
Learning phase	Random walk	Random Walk
Extending pictures sequences	yes	yes
Can it be in the middle	yes	yes
Navigation	yes	Yes
Distance estimation	yes	yes

Note: “Yes” and “No” refer to the inclusion of a task in an experimental block.

Learning Phase: We used different protocols for the learning phases of experimental blocks as follows:

- 1) In the “Random walk” protocol participants learned associations between graph nodes by observing a sequence of pairs of pictures which were sampled from a random walk on the graph (successive pairs of pictures shared a common picture). Participants were instructed to ‘say something in their head’ in order to remember the associations. Hexagonal graphs included 120 steps of the random walk per block and community structured graphs included 180 steps per block (we introduced more pictures in the community graph condition as random walks on such graphs result in high sampling of transitions within a certain community and low sampling of transitions between communities).
- 2) in the “Pairs” protocol participants learned the associations between graph nodes by observing pairs of pictures. Each pair of pictures corresponds to two neighbouring nodes (i.e., an edge) on the graph. Some edges were excluded from the graph (“missing links”), otherwise, the pairs were sampled uniformly randomly according to a uniform distribution and independently across pairs. 150 pairs were presented in each block (with repetition).

The reason we used the “pairs” protocol for Day 2 of Exp. 1 is as follows: Exp.1 was designed to test participants’ ability to infer missing graph links (edges). However, a link that is constantly missing may lead to an inference of the existence of an obstacle rather than an unobserved link. We speculated that learning by sampling pairs of neighbouring nodes, instead of learning from pairs that are taken from random walks on the graph, would reduce this risk. Following the same reasoning, we excluded the navigation task (described below) during the second day of that experiment (hexagonal condition only), as navigation necessarily involves walks on the graph (See “Navigation” rows of Tables 1 and 2”).

Extending pictures sequences: Given a target picture, which of two sequences of three pictures can be extended by that picture (a sequence can be extended by a picture only if it

is a neighbour of the last picture in the sequence, the correct answer can be sequence 1/sequence 2/ both sequences): 16 questions per block. (A picture could not appear twice in the same sequence, therefore, if the target picture is already in the sequence the correct answer was necessarily the other sequence).

Can it be in the middle: Determine whether a picture can appear between two other pictures, the answer is yes if and only if the picture is a neighbour of the two other pictures. 16 questions per block.

navigation task: The aim – navigating to a target picture. Participants are informed that they are currently at the picture that appears on the left of the screen. They were asked to choose between two pictures that are associated with that picture or skip and sample again two pictures that are associated with the current picture (skip is counted as a step). On each step participants were instructed to choose a picture that they think has a smaller number of links to the target picture (according to their memory). Following their choice, the chosen picture appeared on the left and two new pictures, that correspond to states that are neighbours of the chosen picture, appear in the middle (Figure 1b). Once a participant selected a neighbour of the target picture, the target picture itself can appear as a picture that can be chosen. The game terminated when either the target was reached or 200 steps were taken (without reaching the target). In the latter case a message ‘too many steps’ was displayed. On the first block, the number of links from the current picture to the target picture was shown on the screen. Participants played three games in each block. The starting distance (number of links) between the starting picture to the target was 2, 3 and 4.

Distance estimation: Which of two pictures has the smallest number of links to a target picture: 45 questions per block.

Statistical values

All questions: p-value = 0.0068, t = 2.54, sd = 0.0547, ci = [0.0124 inf], d = 0.699

Inference questions: p-value = 0.016, t = 2.19, sd = 0.066, ci = [0.0088 inf], d = 0.565

One-tailed ttest, df = 58.

Second experiment (Figure 6)

Response Time (Learning pace): p-value = 0.003, t = 3.19, sd = 0.5, ci = [0.18, 0.83], d = 1.01 (two-tailed, df = 38).

Correct structural prior leads to faster navigation to the target:

Number of steps to the target is two, p-value = 0.005, t = -2.68, sd = 14.58, ci = [-inf,-4.6], d = 0.85.

Number of steps to the target is three, p-value = 0.026, t = -2.02, sd = 14.85, ci = [-inf,-1.48], d = 0.63.

Number of steps to the target is four, p-value = 0.006, t = -2.6, sd = 10.54, ci = [-inf, -3.04], d = 0.82.

One-tailed ttest, df = 38.

Choose connecting nodes:

All answers: p -value = 0.03, $t = 2.25$, $sd = 0.1$, $ci = [0.007, 0.13]$, $d = 0.71$ (two-tailed, $df = 38$.)

Incorrect answer: p -value = 0.006, $t = 2.88$, $sd = 0.1$, $ci = [0.03, 0.16]$, $d = 0.91$ (two-tailed, $df = 38$.)

ci is the confidence interval and d is Cohen d' (effect size).

The two experiments are in some sense symmetrical, really two conditions of a single experiment. I don't care about the fact that they are described as two different experiments, but what seems more important is the lack of symmetry in the analyses of the data. Why weren't the same analyses applied to both experiments? Have I misunderstood something about the setup that would preclude this?

Thanks - we very much agree that the logic of the conditions is symmetrical, but we, at least, do not think that the analyses can be thought of symmetrically.

The hex allows path inference. The clusters, as we tested them, do not. It would have been possible to leave out links within a cluster. We did not do this because we did not think this would be a compelling demonstration of link inference. If we had done this, we could have done the symmetrical analysis, but to our minds, all of the conceptual interest would have been brought by the Hex condition.

*Similarly the clustered graph has connecting nodes that are qualitatively different from other nodes. The existence of these nodes allows an analysis of behaviour that tests whether the structure of the graph is transferred (For example - connecting nodes are preferred even if they take the subject **away** from the target). There is no equivalent to this analysis in the HEX graph.*

If the reviewer has suggestions for how we can perform symmetrical analyses, we are happy to try them.

There's no direct comparison between the model and experimental data. Ideally we would see the same analyses applied to both model and human participants, plotted in the same way. There is already an indication in the figures that the model is much better than people at this task. The authors should address this mismatch.

As described above, we hope to be able to persuade the reviewer that the role that the model currently performs in the manuscript is different to a quantitative model.

In brief, we are not trying to make a model that predicts human behaviour quantitatively. In order to make such a model, we would have to account for the failures of human memory; for the attentional lapses; for the speed of learning. Etc. etc. It would be a highly parameterised model. In our view, such a model would not likely add any conceptual insight over and above the behavioural effects themselves.

Instead, the purpose of our model is illustrative and didactic. It is a model that teaches us what the critical experiment is to perform. It shows that we should test for inference, and transfer. Critically, it shows that inference alone is not enough, and to demonstrate abstraction we need to show transfer.

To highlight this, we have added new modelling to the manuscript, but it is perhaps not the modelling the reviewer was expecting. Instead, we have added new models that do not abstract the graph structure away from the stimuli. Amongst these models, there is one that can do link inference (self-filtered SR), but there are none that can do transfer. That is there are none that would show a difference between groups in our study.

This is the role of the modelling in this manuscript. We understand that this is different from the role of modelling in other manuscripts that are, for example, trying to extract learning rates from human participants. We believe modelling can play both roles, and we hope the reviewer can agree that the role it is playing in this manuscript is valuable.

The main changes are:

Associative representation

Learning such graphs can be accomplished using different types of representations. One solution to such a problem is a conjunctive representation of the stimuli and their relationships; the relationships between the stimuli are encoded by the associations between the representations of the stimuli themselves (figure 2). An example of such representation is the Successor Representation (SR)²⁵. Here the representation of each state (in our setup each stimulus defines a state) encodes the probability to reach any other states in the future. Using such a representation it is possible to determine whether two stimuli are neighbouring nodes on a graph and even to navigate on a graph.

In : “Inferring unobserved trajectories” we have added the following paragraph (beginning of paragraph 3) :

Simple associative models, such as learning the transition matrix between the pictures themselves or learning SR, cannot solve such a problem (Figure 4a). Following Stachenfeld et al², we spectrally filtered the SR that is currently being learnt using its own eigen-decomposition; we reconstructed the SR using the seven most informative eigenvectors only. Such filtration smooth over the unobserved links, which then allows the agent to answer such questions better than chance without the need of knowledge transfer (Figure 4a, $p < 0.001$ one tailed ttest).

In the same section we have added another paragraph at the end:

We then wanted to check whether humans solve such problems using transfer of abstract structural knowledge or whether they exploit a smoothed associative representation. If participants can solve the task without the need of prior knowledge it will imply that associative learning is enough, while if their performance depends on prior knowledge, we can conclude that humans do represent and transfer abstract structural knowledge. To test whether inference of unobserved edges is achieved by humans using clever smoothing of noisy representation or whether an abstract knowledge is being transferred, we designed the following two experiments.

We have changed and spilt figure 4 into:

Figure 4: inference of unobserved links (Hexagonal graph)

- A) Inferring the existence of unobserved edges (links). Left – the task: The agents had to indicate which of two nodes (pictures) has smaller number of links to the target. With only observed links, the number of links to the target was identical. Right – red edges indicates missing links on the graph. For example, the two nodes that are marked with light blue have the same number of observed links to the target node (marked with dark blue circle), while the number of links that connect these two nodes to the target is different on the complete graph.
- B) When learning from pairs that were sampled randomly (not in succession) while some of the links (pairs) were never observed, simple associative models as learning transition matrix (DA) or simple SR (SR-online: learning using TD-SR²⁸, SR-A: calculating SR from the learnt transition matrix) could not infer the existence of the unobserved links and solve the task (it in fact solves it worse than chance). Agents that use a filtered SR representation (SRreg) could answer these questions better than chance. Shadows are the standard errors.
- C) When learning from pairs that were sampled randomly (not in succession) while some of the links (pairs) were never observed), the basis set agent, that transfers abstract structural knowledge, was able to infer the structural form (Figure S2) and graph size correctly (upper panel). Further, the agent was able to infer the existence of links that were never observed and determined correctly, which of two pictures is closer to a target picture, according to the complete graph (green). The agent could do so even though the number of observed links between the two pictures and the target was identical ($p(\text{cor})$ corresponds to the average fraction of correct answers out of 40 questions in each block). When the agent was forced to infer a community structure

(red), it answered these questions worse than chance. Shadows are the standard errors.

Figure 5 Transfer of structural knowledge allows inference of unobserved links (Hexagonal graph)

Participants had to indicate which of two pictures is closer to a target picture. Participants that reach the second day of our task with the correct prior expectation over the structural form performed significantly better in such task compared to participants with the wrong structural prior (left panel). (30 participants in each group). They were able to answer these questions significantly above chance even when there were links that were never observed, and they had to choose between two pictures with identical number of observed links to the target (right panel). Error bar: SEM. Colorcode: $\text{Log}_{10}(p_{\text{value}})$

There are no comparisons with alternative models that could conceivably solve the same task. For example, Kemp & Tenenbaum (2008) propose a different (and much richer) generative model for graphs. Or within the class of models considered in this paper, what about alternative basis sets? Does the particular choice of basis set made here predict a distinctive pattern of generalization (compared to other choices)? It's also worth pointing out that the authors restricted the structural forms to only the classes that appeared in their experiment, whereas more generally the space is presumably much broader (as argued by Kemp & Tenenbaum).

Thanks again. This is a similar point to the last. We totally agree that the Kemp and Tenenbaum models will perform the same way **because they abstract the graph structure away from the stimuli**. We hope that we have now been clear that the point of the model is to demonstrate the empirical consequences of this abstraction. The model is not an attempt to give detailed insight into the true mechanisms underlying human behaviour.

Given that for the purpose at hand, the K&T model is the same as ours, we hope the reviewer agrees that it is not necessary to re-implement the model. We have now made sure it is clearly stated in the manuscript that the K&T model would perform similarly.

We have added the following sentences to the discussions:

Generalizing structural knowledge in the form of structural forms have been suggested previously by Kemp et al. They have suggested a generative model for constructing a graph using general structural elements, each belonging to a different structural form⁹. As this model exploits abstract structural knowledge, we expect it to perform well on our tasks as well. Our choice of basis sets for representing structural knowledge is inspired both by spectral graph theory^{13,20,32} but mainly by existing research on the hippocampal – entorhinal system.

The authors have done a good job connecting their work to ideas in machine learning and neuroscience, but apart from Tolman there is very little discussion of relevant work in psychology. Below I've given a few references that explore relational/causal transfer. I'm not sure all of these citations are needed; I will leave this to the authors' discretion depending on how they want to discuss this research.

- Halford et al. (1998), Cognitive Psychology -> added this citation
- Honey & Watt (1998), JEP:ABP [this work stimulated a line of research on acquired relational equivalence]-> their experiments show imply that there is heirachical representation of abstract rule
- Kemp et al. (2010), Cognitive Science – have added this citation
- Lu et al. (2016), Cognitive Science (not sure I understand its relevant)
- A large literature on analogical reasoning. For a review of the early work in this area, see Reeves & Weisberg (1994), Psychological Bulletin.

Thanks for the suggestions – we have added most of them to our citation list.

Minor comments:

p. 3: "a learning phase, following" -> "a learning phase. Following"

Fig 3 caption: "that determined" -> "that is determined"

p. 7: what statistical test is being reported for the data in Fig 4b?

p. 8: why is the t-test one-tailed? Same question for all the other one-tailed tests reported in the paper. Sometimes the tests are two-tailed, with no explanation of why one-tailed vs. two-tailed is chosen. I think the default should be two-tailed.

We tried to stick to the statistical practice of performing 1-tailed t-tests when we were testing for effects that were greater than chance (so that only one effect would have been meaningful) and 2-tailed t-tests in other situations. **However, all of our effects survive 2-tailed t-tests.** If the reviewer prefers that we report these numbers, we can happily do so. We think our current approach makes the hypotheses clearer.

p. 8: how was the high-performing subset of participants identified?

We didn't do any tests that required a sub-division of the group. All of the tests were done on the whole group. However, we noted in the text that the effects were driven by high-performing subjects. But we did not need a threshold or criterion because we did not hypothesize this clustering beforehand and we therefore tested the whole group together.

p. 8: "These effects cannot be driven by non-inferential approximations of graph distance". I don't think the relevant issue is about inference vs. non-inference, but rather about the generalization abilities of different models. If you parametrized a successor representation with a basis set that can generalize across graphs, then I see no reason why it couldn't in principle do this task.

This sentence refers to the difference between the groups. Difference between the groups imply transfer. We agree that using basis sets to parametrise SR should work as well, if the basis sets are already known. Actually, if we need to gamble what is happening in the hippocampus formation, we will gamble on that option. The point of our model was to introduce the basis sets as transferable representation of the abstract structural knowledge.

p. 16: "inferece" -> "inference"

p. 17: "a basis sets" -> "a basis set"

p. 17: "using standard interpolating method" -> "using a standard interpolating method" [I actually don't know what standard interpolation method applies to eigenvector resizing]

We added the method (matlab imresize)

p. 17: "Participnats" -> "Participants"

p. 17: "taking from" -> "taken from"

p. 17: why is "bank" capitalized?

p. 18: "Madaresz" -> "Madarasz"

We thank the reviewer on all these comments.

Reviewers' Comments:

Reviewer #1:

Remarks to the Author:

The authors have revised the manuscript and it is now substantially improved. I have no further comments.

Reviewer #2:

Remarks to the Author:

I'm happy with the changes that the authors have made in response to my comments. Just a few minor things:

"massage" -> "message"

How can some of the confidence intervals be bounded by infinity?

"Such filtration smooth" -> "Such filtration smooths"

Please see below our answers to reviewer 2 comments:

1. Massage on page 23 was changed to message
2. Smooth on page 9 was changed to smooths
3. Confidence interval: the confidence intervals are bounded by infinity when we used one tailed ttest. Therefore, it means that one side of the distribution is entirely in the non-rejecting area of the NULL. We used one tailed ttest when we had a particular hypothesis regarding the sign of the effect.

We thanks the reviewers for the comments on our manuscript.

Thanks
Shirley Mark